# MovieDreamer: Hierarchical Generation for Coherent Long Visual Sequences

**Canyu Zhao**[1,*]   **Mingyu Liu**[1,*]   **Wen Wang**[1]   **Weihua Chen**[2]   **Fan Wang**[2]
**Hao Chen**[1]   **Bo Zhang**[1]   **Chunhua Shen**[1]
[1]Zhejiang University
[2]Alibaba Group

## Abstract

Recent advancements in video generation have primarily leveraged diffusion models for short-duration content. However, these approaches often fall short in modeling complex narratives and maintaining character consistency over extended periods, which is essential for long-form video production like movies. We propose MovieDreamer, a novel hierarchical framework that integrates the strengths of autoregressive models with diffusion-based rendering to pioneer long-duration video generation with intricate plot progressions and high visual fidelity. Our approach utilizes autoregressive models for global narrative coherence, predicting sequences of visual tokens that are subsequently transformed into high-quality video frames through diffusion rendering. This method is akin to traditional movie production processes, where complex stories are factorized down into manageable scene capturing. Further, we employ a multimodal script that enriches scene descriptions with detailed character information and visual style, enhancing continuity and character identity across scenes. We present extensive experiments across various movie genres, demonstrating that our approach not only achieves superior visual and narrative quality but also effectively extends the duration of generated content significantly beyond current capabilities.

## 1 Introduction

Driven by the advent of generative modeling techniques (Rombach et al., 2022; Podell et al., 2023), there has been significant progress in video generation in the research community (Guo et al., 2024; Ho et al., 2022b; Blattmann et al., 2023). Tremendous research efforts have focused on adapting a pre-trained text-to-image diffusion model (Guo et al., 2024; Ho et al., 2022b) to a video generation model. Recently revolutionary leap has been made with the Sora (Brooks et al., 2024) model which substantially scales a spatial-temporal transformer model. This model demonstrates remarkable video generation quality, dramatically extending the duration of generated videos from seconds to an unprecedented full minute. This technology profoundly reshapes the boundaries of generative AI, fostering anticipation towards hours-long movie production.

While predominant video generation methods adopt diffusion models, this paradigm is more suited for visual rendering and is less adept at modeling complex abstract logic and reasoning compared to autoregressive models, as evidenced in natural language processing. Moreover, diffusion models lack the flexibility to support arbitrary length. On the other hand, autoregressive models (Wang et al., 2023a; Jiang et al., 2022; Hu et al., 2023; Pan et al., 2023) have shown superior ability in handling complex reasoning and are better at predicting what might happen in the next—a key element of a world model. Moreover, autoregressive models offer greater flexibility in handling varied lengths of predictions and can also benefit from mature training and inference infrastructure. As such, there are attempts that use the autoregressive model for video generation (Kondratyuk et al., 2023; Hu et al., 2023). However, autoregressive modeling is not as compute efficient as diffusion model for visual rendering and demands substantially more computing even for image rendering.

Recent progress (Henschel et al., 2024; Chen et al., 2023; Qiu et al., 2023; Brooks et al., 2024) suggests that video generation quality in short duration (from seconds to a minute) can be consistently improved by scaling the compute due to the scaling law. In this work, we sidestep the short video generation problem and instead seek a generative framework for long video generation featuring

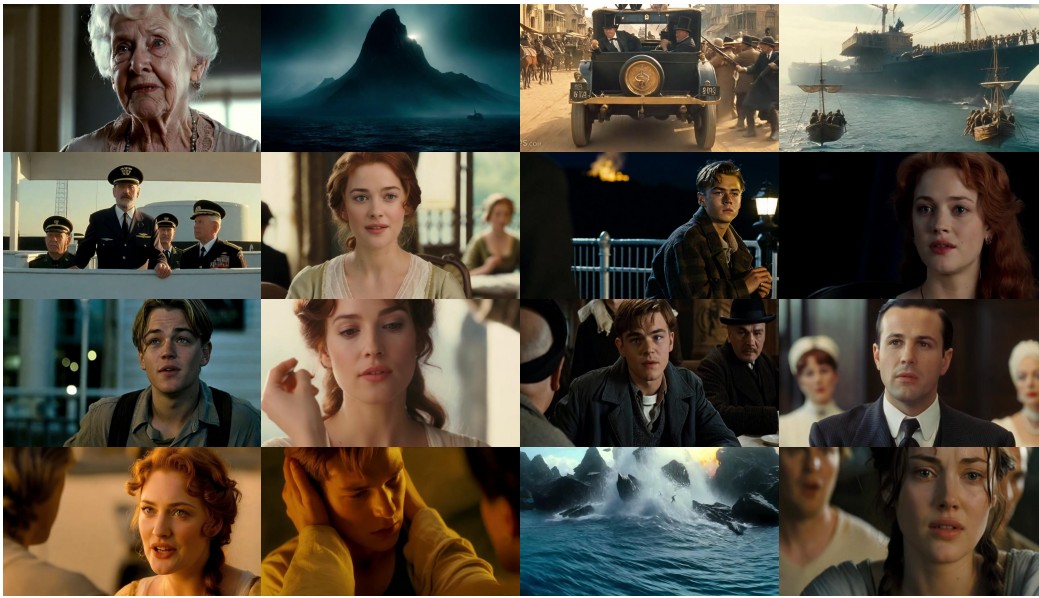

Figure 1: The Titanic movie scenes generated by the proposed MovieDreamer. The generation is conditioned on the elaborate multimodal movie script. *The original movie is unseen during training.* **Please refer to Figure 10, 11, 12, 13, 14 for more results.**

complex narrative structures and intricate plot progressions, which is hardly achieved by merely scaling the current approaches. Our aim is to push the boundaries of video generation beyond the limitations of short-duration content, enabling the creation of videos with rich, engaging storylines.

In this paper, we introduce a hierarchical approach, dubbed *MovieDreamer*, that marries autoregressive modeling with diffusion-based rendering to achieve a novel synthesis of long-term coherence and short-term fidelity in visual storytelling. Our method leverages the strengths of autoregressive models to ensure global consistency in key movie elements like character identities and cinematic styles, even when the camera switches back and forth. The model predicts visual tokens for specific local timespans, which are then decoded into keyframes and dynamically rendered into video sequences using diffusion rendering. Our method mirrors the traditional movie directing process, where intricate plots are decomposed into distinct, manageable scenes. Our method ensures that both the overarching narrative and the minute spatial-temporal details are maintained with high fidelity, thus enabling the automated production of movies that are both visually cohesive and contextually rich.

To this end, we represent the whole movie using sparse keyframes and employ a diffusion autoencoder to tokenize each keyframe into compact visual tokens. We train an autoregressive model to predict the sequence of visual tokens. This model is initialized from a pretrained large language mode, a 7B LLaMA model (Touvron et al., 2023), to leverage its rich world knowledge for better generalizability. As such, the model can be regarded as a multimodal model, which, rather than outputting the text, predicts the visual tokens with movie script conditioning.

Three key designs need to be specially considered during the autoregressive training. One is to resort to various model and input perturbation techniques to combat the overfitting tendency since the high-quality movie training data is limited. Besides, we propose to condition the model using a novel *multimodal script*, which, for each keyframe, includes the rich description of the image style and scene elements, along with detailed text description of characters as well as their retrieved face embeddings. Such multimodal script facilitates narrative continuity across different segments of the movie while offering flexibility for character control. Moreover, we stochastically append a few reference frames of the same episode at the beginning of the input, and in this way, the model acquires in-context learning capability and can synthesize higher-quality personalized results. Such feature is particularly useful in generating coherent continuations of given movie episodes.

We subsequently decode the predicted vision tokens using an image diffusion decoder and further render the image to the video clip using an image-to-video diffusion model. Importantly, we enhance

this process by finetuning an *identity-preserving* diffusion decoder. This refinement effectively improves identity preservation in the resultant video clips. Our work is essentially orthogonal to the current efforts aimed at improving the short video generation quality via compute scaling and our method can benefit from these progresses. Our approach is validated through extensive experiments on a wide range of movie genres, showcasing an excelling ability to generate visually stunning and coherent long-form videos over the state-of-the-art.

To summarize, the contributions of this paper are:

- We introduce MovieDreamer, a novel hierarchical framework that marries autoregressive models with diffusion rendering to generate long-term coherent stories and videos with multiple characters well-preserved. The method substantially extends the duration of generated video content to thousands of keyframes. *To the best of our knowledge, we are the **first** to tackle the ultra-long narrative video generation problem, rather than merely clip looping.*

- We generate the visual token sequence using a multimodal autoregressive model. The autoregressive model supports zero-shot and few-shot personalized generation scenarios and supports variable length of keyframe prediction.

- We use a novel multimodal script to hierarchically structure rich descriptions of scenes, plots, as well as the character's identity. This approach not only facilitates narrative continuity across different segments of a video but also enhances character control and identity preservation.

- Our method demonstrates superior generation quality with detailed visual continuity, high-fidelity visual details, and the character's identity-preserving ability. Both story generation and video generation can be addressed with our method.

## 2 RELATED WORK

**Video generation models.** Video generation has advanced mainly through diffusion and autoregression methods. The popularization of Stable Diffusion (Rombach et al., 2022) has sparked extensive studies, improving video quality with various techniques (Guo et al., 2024; Ho et al., 2022b;a; He et al., 2023). For long videos, strategies like (Yin et al., 2023; Qiu et al., 2023; Wang et al., 2023b; Chen et al., 2023; Henschel et al., 2024) can only generate simple results. Despite their high computational load and challenges in managing complex sequences, diffusion models are widely used. Autoregression, on the other hand, is less common but excels in specific areas like autonomous driving by employing unified token spaces (Hu et al., 2023), although it struggles with continuity in complex scenes. Our approach combines the strengths of both methods in a hierarchical and scalable manner, enhancing control and flexibility irrespective of video length.

**Multimodal large language models.** Recent advancements in NLP lead to the emergence of visual language models (VLMs) through enabling large language model (Touvron et al., 2023) to comprehend visual modalities (Liu et al., 2023b; Bai et al., 2023a;b; Chen et al., 2024). In addition to language, VLMs have demonstrated their power in generating images (Pan et al., 2023; Sun et al., 2023c;b). Moreover, ongoing research (Zhang et al., 2023; Lin et al., 2023; Li et al., 2023a; Zhao et al., 2022) has showcased the robust capabilities of Multimodal Language Models (MLLMs) in comprehending video content. Some works (Cai et al., 2024; Cheng et al., 2024; Ren et al., 2024b; Rasheed et al., 2024; Lai et al., 2024) enable VLMs to assist in spatial understanding and other visual tasks. VLMs (Jiang et al., 2022; Kim et al., 2024; Etukuru et al., 2024; Brohan et al., 2023; Driess et al., 2023) also demonstrate the powerful capabilities in robotics. These advancements have solidified the effectiveness of VLMs in addressing complex problems. Building on this foundation, our approach leverages VLMs to facilitate the generation of extremely long video content.

**Visual story generation.** The crux of visual story generation lies in ensuring consistency across generated images. Similar to video generation, diffusion and autoregression are also two primary manners. Existing diffusion-based methods predominantly employ conditional generation to maintain the coherence (Su et al., 2023; Wang et al., 2023c; Tewel et al., 2024; Hertz et al., 2023). Some (Li et al., 2023b; Wu et al., 2024) have shown promising results in retaining high-quality facial features and identity. Other approaches are based on the autoregressive ideas (Feng et al., 2023; Liu et al., 2023a; Rahman et al., 2023; Pan et al., 2022). However, these approaches generally yield

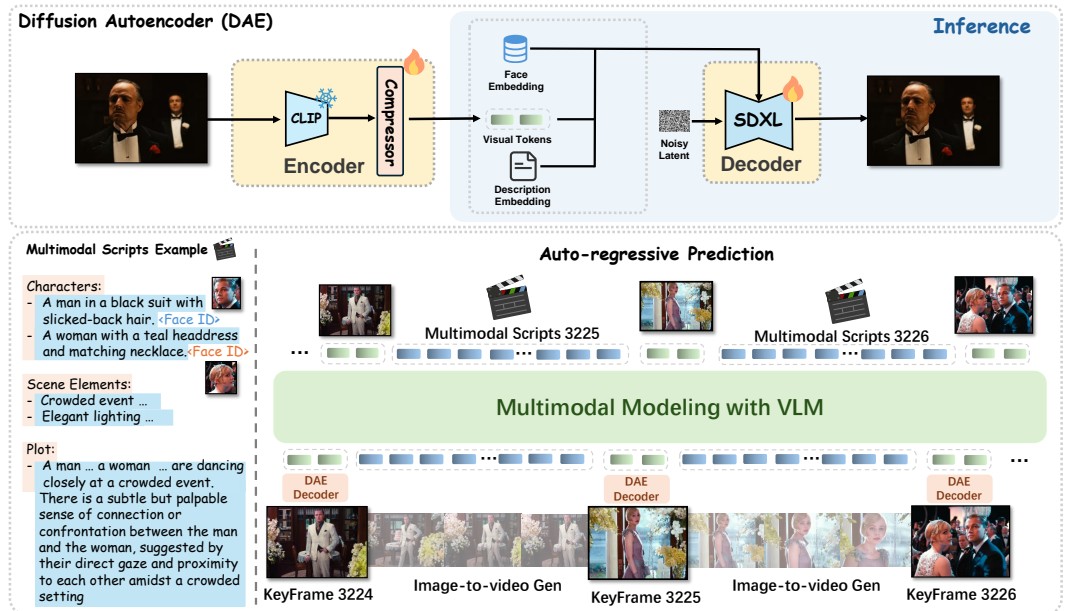

Figure 2: The framework of our MovieDreamer. Our autoregressive model takes multimodal scripts as input and predicts the tokens for keyframes. These tokens are then rendered into images, forming anchor frames for extended video generation. Our method ensures long-term coherence and short-term fidelity in visual storytelling with the character's identity well preserved.

mediocre results and are challenging to scale up. Story and video generation share intrinsic links, with overlapping solutions. Our method uniquely integrates these, offering a unified approach.

## 3 METHOD

### 3.1 OVERVIEW

We propose a novel framework for generating extended video sequences that leverages the strengths of autoregressive models for long-term temporal consistency and diffusion models for high-quality image and video rendering. Our method takes multimodal scripts as conditions, predicts keyframe tokens in an autoregressive manner, renders the tokens into images and uses these images as anchors to produce full-length videos. Our method offers flexibility to support zero-shot generation along with few-shot scenarios in which the generation results are required to follow the given reference. We take special care of preserving the identity of the characters throughout the multimodal script design, autoregressive training, and diffusion rendering. We illustrate the overall framework in Figure 2.

### 3.2 KEYFRAME TOKENIZATION VIA DIFFUSION AUTOENCODER

To create a concise yet faithful representation of images, we employ a diffusion autoencoder. Our encoder, $\mathcal{E}$, contains a pretrained CLIP vision model (Radford et al., 2021) and a transformer-based token compressor, which encodes an image $\mathbf{x}$ into a reduced number of compressed tokens denoted by $\mathbf{e}$. The decoder, $\mathcal{D}$, is finetuned based on the pretrained SDXL and yields an $896 \times 512$ reconstructed image by leveraging the compressed tokens in the cross-attention module. We train the compressor and the decoder while leaving the CLIP vision model frozen. The loss for training this diffusion autoencoder is as follows:

$$\mathcal{L}_{DAE} = \mathbb{E}_{\mathbf{x}_0, \epsilon \sim \mathcal{N}(0, \mathbf{I})} \|\epsilon - \mathcal{D}(\mathbf{z}_t, t, \mathcal{E}(\mathbf{x}_0))\|_2^2, \tag{1}$$

where $\mathbf{x}_0$ is the input image and $\mathbf{z}_t = \alpha_t \mathbf{x} + \sigma_t \epsilon$ is its corresponding noisy latent at timestep $t$. $\epsilon \in \mathcal{N}(0, I)$ is the Gaussian noise. $\alpha_t$ and $\sigma_t$ define the noise schedule. In our experiment, we

empirically find that merely two tokens can sufficiently characterize major semantics of keyframes, corroborating the finding of previous work (Yang et al., 2022).

### 3.3 ID-PRESERVING DIFFUSION RENDERING

In our approach, while the primary diffusion decoder $\mathcal{D}$ adeptly reconstructs target images, it occasionally falls short in capturing fine-grained details, notably in facial features, due to the attenuation of details in compressed tokens. To address this, we enhance the cross-attention module within $\mathcal{D}$. This enhancement involves the integration of both descriptive text embedding, $\mathbf{d}$, and face embedding, $\mathbf{f}$, derived from the multimodal script. Specifically, the face embedding $\mathbf{f}$ and description embedding $\mathbf{d}$ is projected to the same dimension as the compressed token with two additional MLPs, which are then concatenated together and serve as the input to the keys and values of cross-attention modules in $\mathcal{D}$. The training loss is:

$$\mathcal{L}_{DAE} = \mathbb{E}_{\mathbf{x}_0, \epsilon \sim \mathcal{N}(0, \mathbf{I})} \|\epsilon - \mathcal{D}(\mathbf{z}_t, t, \mathcal{E}(\mathbf{x}_0), \mathbf{f}, \mathbf{d})\|_2^2, \tag{2}$$

To further advocate the model's ability to focus on critical details, we introduce a random masking strategy that obscures a subset of the input tokens as zero tokens only during training. This technique encourages the decoder to more effectively utilize the available facial and textual cues to reconstruct the images with higher fidelity, particularly in preserving identity-specific characteristics. This ID-preserving rendering also compensates for the loss of identity during the autoregressive modeling, leading to substantially improved identity perception quality as illustrated in Figure 3.

### 3.4 AUTOREGRESSIVE KEYFRAME TOKEN GENERATION

We initiate our approach by leveraging a pretrained large language model, specifically LLaMA2-7B (Touvron et al., 2023), to construct our autoregressive model, $\mathcal{G}$. Unlike traditional LLMs, $\mathcal{G}$ is designed to predict compressed vision tokens from the multimodal script $\mathbf{M}$ of the corresponding keyframe and historical information $\mathbf{H}$, which is formulated as $\mathbf{e}_t = \mathcal{G}(\mathbf{H}_{<t}, \mathbf{M}_t)$.

Traditional LLMs often employ cross-entropy loss for training, which is suitable for discrete outputs. However, our model deals with continuous real-valued image tokens, which makes cross-entropy inapplicable. Inspired by Tschannen et al. (2024), we adopt a $k$-mixture Gaussian Mixture Model (GMM) to effectively model the distribution of these real-valued tokens. This involves parameterizing the GMM with $kd$ means, $kd$ variances, and $k$ mixing coefficients. These parameters are predicted by attaching an additional trainable linear layer at the end of the autoregressive model, enabling the construction of the GMM and the sampling of continuous tokens from the GMM. The model is trained by minimizing the negative log-likelihood:

$$\mathcal{L}_{nll} = \mathbb{E}(-\sum_{t=1}^{T} \sum_{i=1}^{L} \log p(e_{t,i}|\mathbf{H}_{<t}, \mathbf{M}_t)), \tag{3}$$

where $T$ represents the total number of frames, $t$ indexes the current frame, and $L$ is the number of compressed tokens per frame. To ease the learning for these continuous tokens, we also incorporate $\ell_1$ and $\ell_2$ losses between the predicted and ground truth token and optimize the following objective:

$$\mathcal{L}_{ar} = \mathcal{L}_{nll} + \ell_1(\mathbf{e}_{pred}, \mathbf{e}_{gt}) + \ell_2(\mathbf{e}_{pred}, \mathbf{e}_{gt}). \tag{4}$$

**Anti-overfitting.** It is challenging to massively acquire large amounts of high-quality, long-duration video data. We managed to collect 50k long videos, yielding 5M keyframes for autoregressive training. Yet this data volume is relatively limited, leading to a severe overfitting issue where the generative model fails to generalize well during testing. To address this, we implement several strategies which are crucial for training:

- Data augmentation. To maximally utilize our training data, we apply random horizontal flips and stochastically reverse the temporal order of video frames. Such training data augmentation considerably increases the diversity of training data.

- Face embedding randomization. To prevent identity leakage, we randomly retrieve face embeddings of the same character from different frames. Otherwise, the model will memorize the training frame simply from the face embedding input.

- Aggressive dropout. An unusually high dropout rate of 50% is utilized, which is crucial for generalized learning from limited training data.
- Token masking. We incorporate random masking of input tokens with a probability of 0.15, which is applied to the causal attention mask. This forces the model to infer missing information based on the available context (face ID), further strengthening its ability to generalize from partial data.

**Multi-modal scripts for autoregressive conditioning.** We develop a well-structured multi-modal script format to serve as input for autoregressive models, as depicted in Figure 16. Our script integrates multiple dimensions: characters, scene elements, and plots. Accurately representing character appearance using text alone is challenging; therefore, we combine textual descriptions with facial embeddings (Zheng et al., 2022) to provide a more detailed representation of each character. To facilitate the processing by the autoregressive model, we structure the script format to distinctly delineate these elements.

For non-textual modalities such as facial embeddings and compressed tokens are projected to LLaMA's embedding space using multi-layer perceptron. The primary challenge arises with textual data, which tends to produce long sequences, thereby consuming excessive token space and limiting the model's contextual breadth. This challenge poses a significant obstacle to achieving our goal of *long* content generation. To mitigate this, we treat text as a separate modality, dividing it into "identifiers" and "descriptions" (elaborated in Appendix A). Identifiers establish the script's structure, while descriptions detail the attributes for generation. Descriptions are segmented into sentences, with each encoded into a single `[CLS]` token using CLIP and then projected into the unified input space. In other words, we perform the sentence-level tokenization.

This method significantly extends the usable contextual length during training by condensing an entire sentence into a single token. We utilize LongCLIP (Zhang et al., 2024) as our text encoder for descriptions, supporting inputs up to 248 tokens, which enhances our capacity to handle detailed narrative content. Consequently, the multi-modal script at timestep $t$ and preceding historical information are represented as:

$$\mathbf{M}_t = [\mathbf{d}_t, \mathbf{i}_t, \mathbf{c}_t], \ \mathbf{H}_{<t} = [\mathbf{d}_{<t}, \mathbf{i}_{<t}, \mathbf{c}_{<t}, \mathbf{e}_{<t}], \tag{5}$$

where $\mathbf{d}_t$, $\mathbf{i}_t$, and $\mathbf{c}_t$ denote the embeddings for descriptions, identifiers, and characters' facial embeddings, respectively. $\mathbf{e}_{<t}$ represents the previously predicted compressed frame tokens. The negative log-likelihood loss in Equation 3 is then formulated as:

$$L_{nll} = \mathbb{E}(-\sum_{t=1}^{T}\sum_{i=1}^{L}\log p(e_{t,i}|e_{t,<i}, \mathbf{e}_{<t}, \mathbf{c}_{\leq t}, \mathbf{d}_{\leq t}, \mathbf{i}_{\leq t})). \tag{6}$$

**Few-shot training for personalized generation.** To promote personalized movie content generation, we propose a few-shot learning method that utilizes in-context learning. During training, we select 10 random frames from an episode, encode them into visual tokens, and prepend these tokens stochastically to the input. This strategy not only promotes in-context learning, allowing the model to tailor content based on the reference frames, but also acts as a data augmentation technique, effectively mitigating overfitting.

Our model is versatile, supporting both zero-shot and few-shot generation modes. In zero-shot mode, the model generates content from the multimodal script. In few-shot mode, it leverages a small set of user-provided reference images to align the generated content more closely with user preferences, without necessitating further training. This capability ensures that users can efficiently produce high-quality, customized visual content that aligns better with their desired target.

## 3.5 KEYFRAME-BASED VIDEO GENERATION

After obtaining the keyframes, we can generate video clips of the movie based on these keyframes. A straightforward approach is to utilize an existing image-to-video model, such as Stable Video Diffusion (SVD) (Blattmann et al., 2023), to generate these clips. Specifically, SVD transforms the input image into latent features for conditioning and introduces interaction with the CLIP features of the input image via cross-attention. Although SVD is capable of generating high-quality short videos, *e.g.*, 25 frames, it struggles to generate longer movie clips.

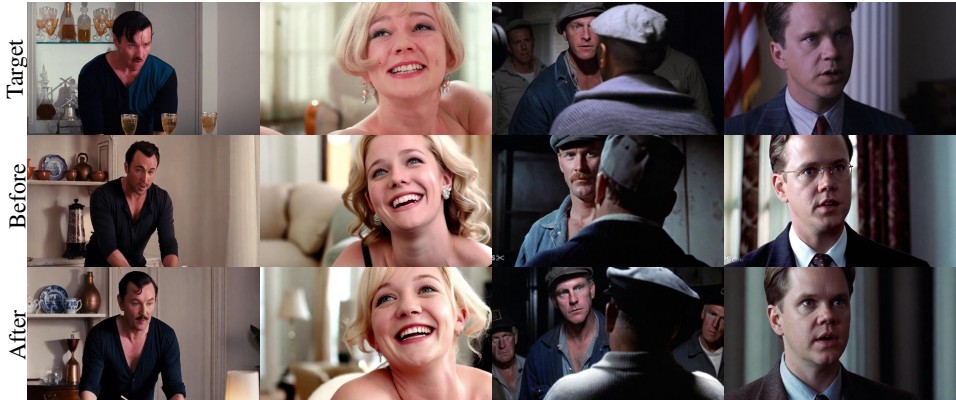

Figure 3: The ID-preserving renderer significantly enhances the perceived identity of the character. For the second row, the input is only the compressed token. For the third row, the input is the compressed token, face ID, and description embedding.

To generate longer movie clips, a straightforward way is to utilize the last frame of the previously generated video as the initial frame for generating the subsequent video. This process can be iterated to obtain a lengthy video sequence. However, we empirically found that this causes a serious accumulation of errors: the quality of the video frames gradually deteriorates as the time gets longer.

To tackle this, we propose a simple and effective solution. Our motivation is to always use the feature of the first frame as an "anchor" during video extension, to enhance the model's awareness of the original image distribution. In practice, we use the CLIP feature of the original input image, instead of the last frame of the previous video, for cross-attn interaction when generating subsequent videos.

*We want to emphasize that our key contribution lies in the **hierarchical** controllable very long video generation, with MLLM for keyframe generation and diffusion model for video rendering.* Our subsequent experiments also demonstrate that existing high-quality video generation models trained by resource-rich institutions can be integrated with our MovieDreamer, producing better visually appealing results. This further demonstrates our flexibility.

## 4 EXPERIMENTS

### 4.1 IMPLEMENTATION DETAILS

We use pretrained models to reduce the training computational overhead, which is very common in the era of large models. We use the U-Net of pretrained SDXL (Podell et al., 2023) as our diffusion decoder. The encoder consists of a pretrained CLIP image encoder (ViT-L) (Radford et al., 2021) and a transformer-based compressor. The autoregressive model is initialized with a pretrained LLaMA-7B (Touvron et al., 2023) checkpoint. All MLPs used to align different modal inputs follow the architecture in LLaVA (Liu et al., 2023b). Please refer to Appendix A for more details about the network architecture and training.

### 4.2 EXPERIMENTAL SETUP

**Dataset.**    The majority of our training data is sourced from Condensed Movies (Bain et al., 2020), with additional data collected using the same methodology as this dataset, resulting in 5M keyframes with corresponding script annotations. To systematically evaluate the effectiveness of our method, we construct a test dataset consisting of 100 long movies that are NOT included in the training set, with 1M keyframes after pre-processing. Each of these videos is processed through our proposed data pre-processing pipeline (see Appendix C), which accurately extracts and annotates keyframes.

**Evaluation metrics.**    We evaluate the effectiveness of various methods using the following metrics. The CLIP score (Radford et al., 2021) assesses semantic alignment between the plot and generated outputs. Quality is evaluated using the Aesthetic Score (AS) (Schuhmann et al., 2022), Fréchet

Image Distance (FID) (Heusel et al., 2018), and Inception Score (IS) (Salimans et al., 2016). For assessing consistency, we propose new metrics for short-term (ST) and long-term (LT) consistency. ST is measured by calculating the CLIP cosine similarity between adjacent frames containing the target character. LT consistency is evaluated by checking the presence of the target character in selected frames across the test set. Finally, we calculate the ratio of frames containing the target character to the total number of selected frames as a measure of the accuracy in maintaining long-term character consistency. Please refer to the Appendix B for more details about the evaluation metrics.

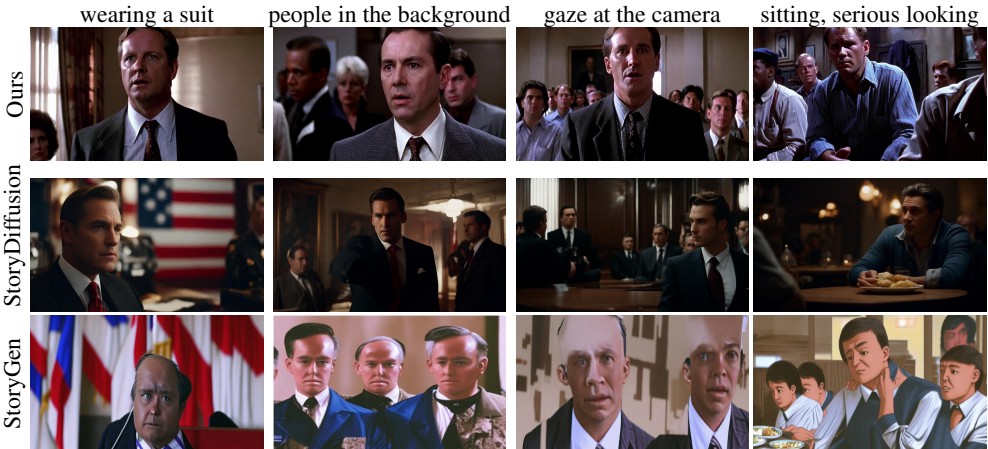

Figure 4: Qualitative comparisons of story generation. Please refer to the appendix E for more comparisons. Each keyframe is generated with the corresponding multimodal script as input.

## 4.3 COMPARISON WITH THE STATE-OF-THE-ART

**Story generation.** Many existing story generation methods focus on fine-tuning with small datasets, exhibiting poor generalization. Consequently, we compare our method only with those that demonstrate high generalization capabilities, namely StoryDiffusion (Zhou et al., 2024) and StoryGen (Liu et al., 2023a). As shown in Figure 4, StoryDiffusion consumes significant computational overhead and tends to generate generic faces that do not align with the desired character. Similarly, StoryGen fails to preserve consistency and generates abnormal results. In contrast, our method achieves generating extremely long content while preserving both short-term and long-term consistency across multiple characters. The observation is also evidenced by the quantitative results in Table 1, where our method achieves high scores in both LT and ST. Moreover, the higher CLIP score reflects that our generated results align well with the storyline. Better IS, AS, and FID scores demonstrate our method achieves high-quality images. We showcase more qualitative results in Appendix E.

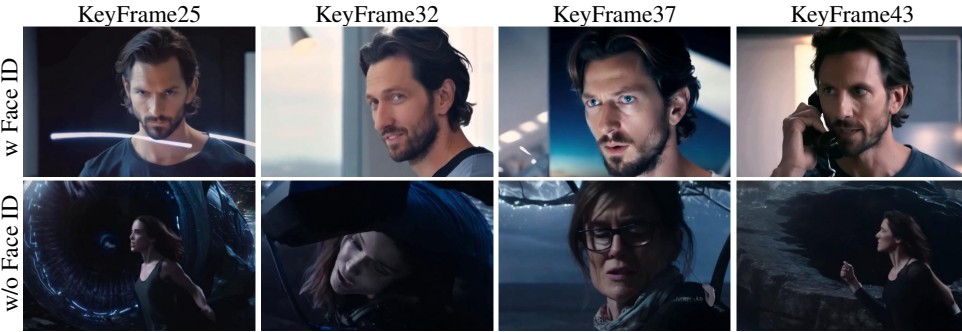

Figure 5: Ablations on face embedding. Character consistency is well-preserved with face embedding. The autoregressive model generates compressed tokens with multimodal script as input. The results are rendered by the decoder $\mathcal{D}$, with compressed tokens as well as face embedding and description embedding as input.

Table 1: Quantitative comparisons of different methods. ST and LT refer to our proposed short-term and long-term consistency metrics, respectively. noC denotes without continuous token supervision.

| Method | CLIP ↑ | Inception↑ | Aesthetic↑ | FID↓ | ST↑ | LT↑ |
|---|---|---|---|---|---|---|
| StoryDiffusion | 15.232 | 8.739 | 6.134 | 4.643 | 0.627 | 0.596 |
| StoryGen | 13.261 | 6.216 | 3.837 | 8.557 | 0.508 | 0.542 |
| Ours | 19.584 | 9.698 | 6.093 | 2.043 | 0.646 | 0.814 |
| Ours-ref | **20.071** | **9.842** | **6.288** | **1.912** | **0.701** | **0.893** |
| Ours-noC | 18.833 | 8.609 | 5.893 | 2.714 | 0.606 | 0.755 |
| Ours-noC-ref | 19.431 | 8.774 | 5.979 | 2.560 | 0.616 | 0.776 |

Table 2: Quantitative comparison on full-length video generation.

| Method | CLIP↑ | Inception↑ | Aesthetic↑ | CLIP-sim↑ |
|---|---|---|---|---|
| StreamingT2V | 15.304 | 7.636 | 4.119 | 0.624 |
| SEINE | 19.404 | 7.463 | 5.825 | 0.688 |
| FreeNoise | 14.327 | 6.532 | 3.194 | 0.612 |
| Ours | **19.520** | **8.642** | **6.049** | **0.704** |

**Video results.** We perform a detailed comparison between our method and existing methods for generating long videos. For text-to-video methods, we use detailed descriptions prepared in the test set as input. For image-to-video approaches, we employ keyframes generated by our methods as inputs. As illustrated in Table 2 and Figure 6 (b), our method significantly outperforms prior open-source models in terms of quality, demonstrating strong generalization capabilities. Most importantly, our method is capable of generating videos that last *tens of minutes or even hours* with little compromise in quality, achieving state-of-the-art results. Here, we compare existing methods with our enhanced video model. As shown in Appendix D, when integrated with enterprise-level closed-source video models, our approach demonstrates even better results.

## 4.4 ANALYSIS

**Continuous token supervision.** We noticed a significant decline in the model's ability to maintain ID consistency on the test set when using only L1 and L2 supervision. We believe this is because L1 and L2 losses are not good at modeling probability distributions, as they are more suitable for regression tasks. However, we need to model the complex conditional distribution of the autoregressive process. Therefore, we applied more refined supervision on continuous real-valued tokens. As illustrated in Figure 6, with continuous token supervision, the model is better capable of preserving target ID. Both the quality of the generated image and the preservation of character ID degenerate without continuous token supervision.

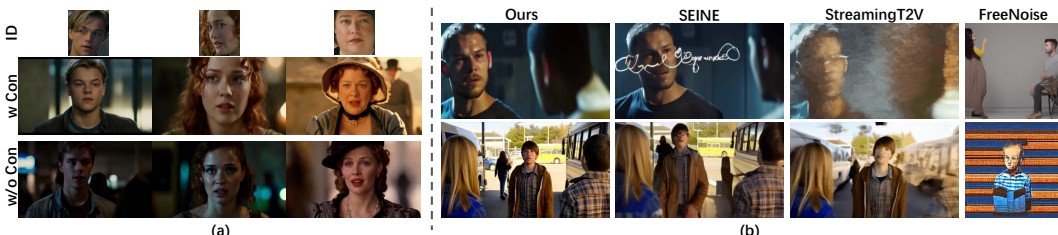

Figure 6: **(a)** With continuous token supervision, our model demonstrates stronger generalization capabilities and achieves higher-quality results. Better character ID is preserved with continuous token supervision. **(b)** Visualization of long video generated by different methods.

**Anti-overfitting strategies.** Large autoregressive models are powerful learners, making it easy for them to overfit the dataset. As shown in the first row in Figure 7, the generated contents are dominated by the input character. The model produces similar visual content even when given different text prompts. Our anti-overfitting strategies are designed to weaken the correspondence between character IDs and target frames, thus avoiding simple memorization. As can be seen in the second row, it helps produce diverse high-quality results that align well with the text description.

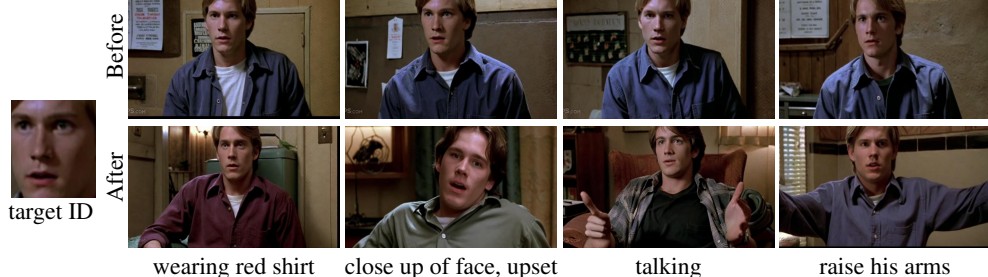

Figure 7: After resolving the overfitting, our model is capable of generating more diverse results. Each keyframe is generated with the corresponding multimodal script as input.

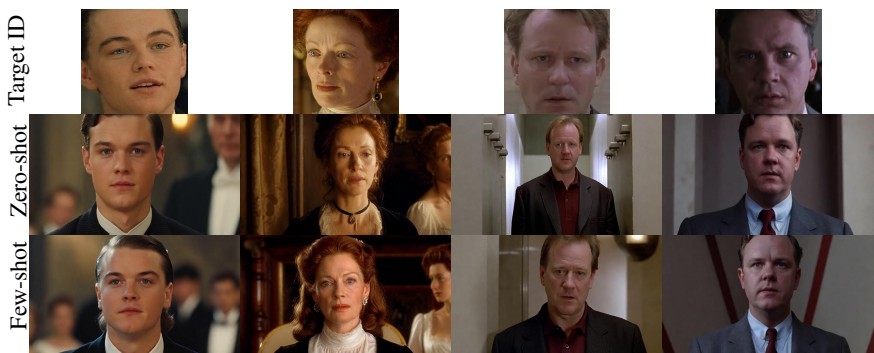

Figure 8: Our zero-shot generation already achieves high success rates and quality. However, it occasionally struggles to preserve ID consistency accurately. Few-shot generation can produce results with better ID preservation.

**Multi-modal movie scripts.**    The multi-modal script introduces face embedding to better preserve consistency. Figure 5 compellingly demonstrates the effectiveness of this design. Specifically, the removal of face embeddings leads to a significantly decreased ability of the model to preserve character consistency. Face embeddings carry more nuanced and precise information compared to text alone. With face embedding, both the short-term and long-term consistency are well-preserved.

**ID-preserving rendering.**    Before enabling ID-preserving rendering, our decoder has already shown the ability to reconstruct the target image. However, for images outside the training set, the reconstructed characters' appearances may differ slightly from the expected targets due to the loss of fine facial features in the compressed tokens. After applying ID-preserving rendering, our decoder exhibits a significantly enhanced ability to preserve character identities. The experimental results, illustrated in Figure 3, clearly demonstrate the effectiveness of the post-processing step.

**Few-shot personalized generation.**    Our method serves as a powerful in-context learner, capable of generating results that are consistent with the style or characters of the few references provided by the user. The results are presented in Figure 8. Our model can produce results that are more consistent with the style and characters in the few-shot scenario.

## 5    CONCLUSION

We present MovieDreamer to address the challenge of generating long-duration visual content with complex narratives. This method elegantly marries the advantages of autoregression and diffusion, and is capable of generating very long videos. Additionally, we design the multi-modal script which aims at preserving character consistency across generated sequences. We further introduce ID-preserving rendering to better preserve character IDs and support few-shot movie creation due to the in-context modeling. This work potentially opens up exciting possibilities for future advancements in automated long-duration video production.

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

# A    IMPLEMENTATION DETAILS

**Diffusion Autoencoder.**    We use the pretrained CLIP image encoder (Radford et al., 2021) to generate spatial tokens. The spatial tokens are compressed into two tokens using the compressor, a network with 6 Transformer blocks. The architecture of the compressor is shown in Figure 9. First, we process the spatial tokens using three transformer blocks. We then interchange the dimensions of the hidden states and compress them using a linear layer. The compressed hidden states are further processed through three transformers to obtain the final compressed token. We find that setting the compressed token to 2 is sufficient for well reconstructing the image.

Subsequently, we employ the pretrained U-Net of SDXL (Podell et al., 2023) as our decoder to render the compressed tokens back into images. Notably, the original SDXL architecture demands an additional projected text embedding as input. To address this, we concatenate and project the compressed token into the desired input space of SDXL simply using a linear layer.

Compared to the image dataset, our collected movie dataset may lack diversity in all kinds of objects in the world, which somewhat limits the model's performance. To ensure the model develops a general understanding of various objects first, we employed a three-stage training strategy to train a more effective model. In the first stage, we fine-tuned the pretrained U-Net and the compressor on the large existing image dataset at a resolution of 768 x 768 with a cosine scheduling learning rate that ranges from 1e-4 to 2e-5 and an AdamW optimizer for 120k steps, allowing the model to understand various objects. Our dataset comprises images from OpenImages (Kuznetsova et al., 2018), JourneyDB (Sun et al., 2023a), and Object365 (Shao et al., 2019).

In the second stage, we further fine-tuned the model using our self-collected movie dataset at a resolution of 896 x 512. The model is trained using AdamW with a constant learning rate of 2e-5 for 120k steps in the second stage. For the first stage and second stage, we first freeze the decoder and only train the compressor for the first 20k steps, driving it to compress images more appropriately. To enable the model to generate results consistent with movie lighting, we further set the noise offset to 0.05 during training.

Figure 9: Architecture of the compressor.

In the third stage, to better preserve character ID, we freeze the compressor and concatenate the compressed tokens with the text embedding and face embedding as inputs to the cross-attention and further finetune the decoder. Text embedding and face embedding are available in multimodal script data. Since the dimensions of the description embedding and facial embedding differ from those of the compressed token, we first project them to the same dimension as the compressed token using 2 two-layer MLPs with GELU as the activation function. Given that different frames have varying numbers of descriptions and face IDs, we pad the projected description embedding and projected facial embedding with zero tensors to extend the sequence length to 15 and 5, respectively. For instance, if a frame has a projected description embedding of shape [12, D] and a projected facial embedding of shape [3, D], we pad them with zero tensors of shape [3, D] and [2, D] to achieve shapes of [15, D] and [5, D]. Finally, we concatenate the compressed tokens with the projected description embedding and projected facial embedding, resulting in a [22, D] tensor, which serves as the key and value input for the UNet cross-attention block. The decoder and the MLPs are trained using AdamW with a constant learning rate of 2e-5 for 120k steps and a noise offset of 0.05. The entire training process of the compressor and decoder takes 3 weeks with 6 NVIDIA A800 GPUs.

**Autoregressive Model.** We utilize a pretrained LLaMA 7B model as the backbone for our autoregressive model, which is subsequently trained with our curated multi-modal script data. We use two-layer MLPs with GELU as the activation function to unify different modalities into the input space of the large language model. We have three distinct modalities that need to be unified into the input space: compressed tokens, facial embeddings, and description embeddings. Therefore, three MLPs are trained alongside the autoregressive model. We set the length of the context frames to 128, which results in the max sequence length around 5000. The global autoregressive model is trained for 3 days using 4 NVIDIA H100 GPUs with a constant learning rate of 2e-5 and 2k steps for warm-up.

To help the model distinguish the script for different frames, we inserted two special tokens, [START_OF_FRAME] and [END_OF_FRAME], at the beginning and end of each frame's script, respectively. Furthermore, in our setting, there are three different modalities: face embedding, compressed tokens, and description embedding. The face embedding is extracted with FARL (Zheng et al., 2022), and for each frame, we extract the top 5 characters. The compressed tokens are generated by leveraging our trained encoder $\mathcal{E}$, with the target frame as input. The description is divided into different sentences, and each sentence is encoded into one [CLS] token using Long-CLIP (Zhang et al., 2024). We treat the remaining text such as "Characters:" and "Scene Elements:" as identifiers, which are tokenized the same as LLaMA, as shown in Figure 2.

**The identifier and descriptive texts in multimodal script.** Here, we provide an example to help readers understand the multimodal script in detail. Consider the example multimodal script in Figure 2, [*Characters:*], [*Scene Elements:* ], [*Plots:*] and the symbol [-] are identifiers, while [*A man in a black suit...*], [*A woman with a teal headdress...*], [*Crowded event*], [*Elegent Lighting*], [*A man ... a woman ... are dancing closely...*] are text descriptions. As shown in Figure 2, orange parts are identifiers, and blue parts are text descriptions. The identifier structures the input format, aiding the autoregressive model in understanding the different types of content within the script. The description is responsible for providing detailed information about the Character, Scene Elements, and Plot.

**ID-preserving Rendering.** To enhance our model's ability to maintain character consistency, we additionally input the description embedding encoded by LongCLIP (Zhang et al., 2024) and the face embedding extracted by FARL (Zheng et al., 2022) into the trained decoder. Both the text embedding and face embedding are aligned with the decoder's input using two-layer MLPs and then concatenated with the compressed tokens to serve as the input for cross-attention.

We assume a maximum of 15 sentences for description embedding and 5 character IDs. If the inputs do not meet these quantities, we pad them with zeros. Moreover, for the concatenated embedding, we randomly mask each token with a probability of 0.2. This encourages the model to learn the relationship between facial features, text descriptions, and the compressed token.

**Continuous Token Modeling.** The $k$-mixutre GMM models the distribution $p(\mathbf{e}_t|\mathbf{H}_{<t}, \mathbf{M}_t)$, where $\mathbf{e}_t$ is the continuous real-valued image tokens of sequence length $S$ and dimension $d$. $\mathbf{H}_{<t}$ and $\mathbf{M}_t$ refer to the previous history information and the current multi-modal script. The GMM consists of $k$ multivariate Gaussian distribution components that are independent across the channel dimension $d$. To construct the GMM, we replace the logit prediction layer of our autoregressive model initialized using LLaMA with a linear layer with $2kd + k$ dimension for output. Subsequently, the mean $m$ and the variance $v$ of shape $S \times kd$ and mixture weights $w$ of shape $S \times k$ are derived. $m$ consists of a stack of tensors $[m^{(1)}, m^{(2)}, \ldots m^{(k)}]$, where $m_i$ is of shape $S \times d$. Similarly, $v = [v^{(1)}, v^{(2)}, \ldots v^{(k)}]$. The mixture weight is normalized along the channel dimension. Given the ground truth image embedding $\mathbf{e}$, we aim to minimize its negative log-likelihood within the GMM:

$$
\begin{aligned}
-\log(p(\mathbf{e}|\mathbf{H}_{<t}, \mathbf{M}_t)) &= -\log\left(\prod_{l=1}^{S}\left(\sum_{i=1}^{k} w_l^{(i)} \prod_{c=1}^{d} \mathcal{N}(e_{l,c}|m_{l,c}^{(i)}, v_{l,c}^{(i)})\right)\right) \\
&= -\sum_{l=1}^{S}\log\left(\sum_{i=1}^{k} w_l^{(i)} \prod_{c=1}^{d} \mathcal{N}(e_{l,c}|m_{l,c}^{(i)}, v_{l,c}^{(i)})\right),
\end{aligned}
\tag{7}
$$

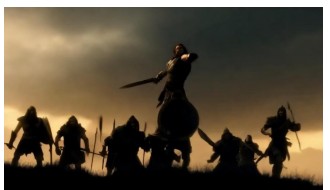

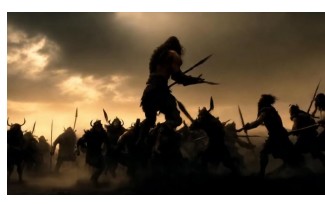

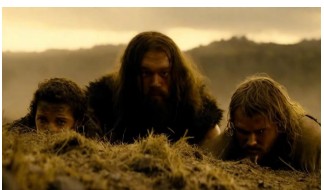

In a distant northern land, there was a tribe renowned for its berserkers, known for their courage and fearlessness.

The berserkers spread war across many lands.

Aron was always at the forefront; he was both the chief and the fiercest warrior.

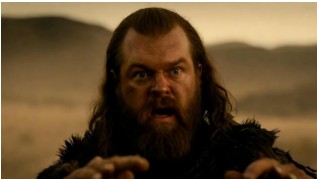

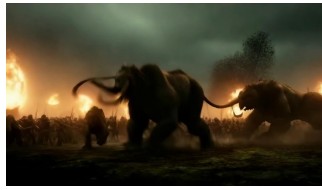

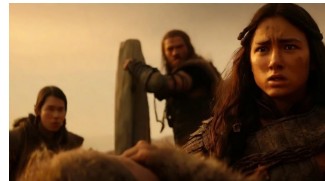

However, one day, Aron discovered that their former victims had united to seek vengeance against their tribe.

The war inflicted immense losses on their tribe.

The tribe members all became aware of the sins they had committed in the past.

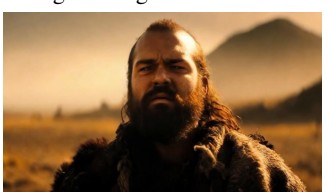

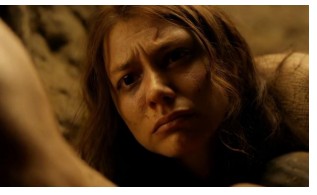

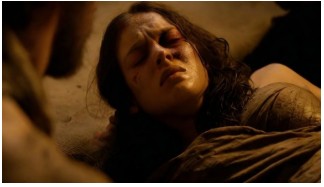

Nevertheless, Aron clung to his previous war strategies.

His wife, Eleanor was injured seriously during the escape. She urged him to see that war was merciless, with no victors—only suffering.

Knowing Aron's stubborn nature, Eleanor lay down in disappointment.

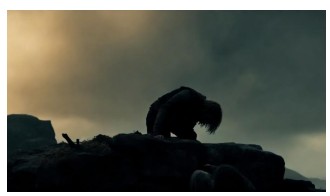

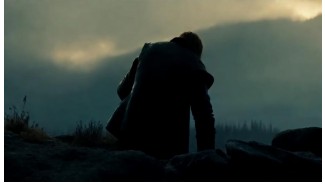

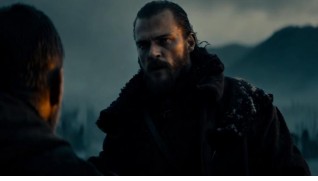

Eleanor's injury and the prolonged suffering eventually made Aron realize that Eleanor was right. He struggled to sleep each night.

One morning, he stared into the distance, deep in thought.

After much reflection, he decided to leave the tribe and embark on a journey of atonement alone.

Figure 10: Additional Zero-Shot results.

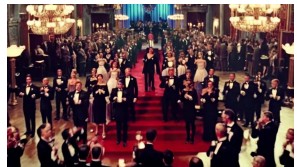

Gatsby often held incredibly lavish parties at his home.

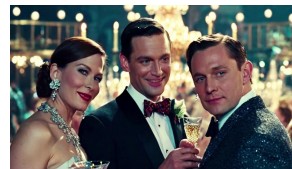

Many famous big shots would attend the parties, but people didn't know who Gatsby really was.

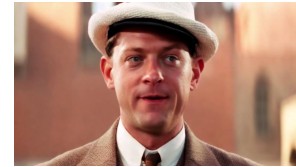

However, Nick received an invitation to Gatsby's party.

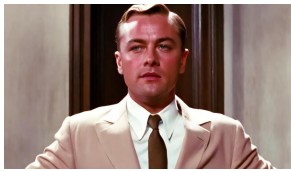

At the party, he met Gatsby, who wanted to rekindle his lost love with Nick's cousin, Daisy.

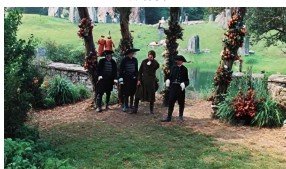

Nick agreed to help Gatsby, so Gatsby had many people meticulously decorate Nick's house.

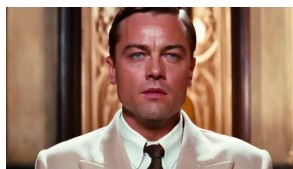

As the appointed time approached, Gatsby became very nervous.

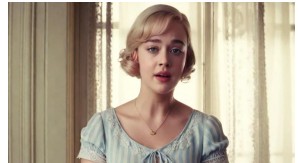

Daisy arrived as promised.

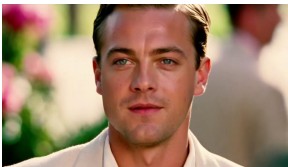

Seeing Daisy, Gatsby's long-suppressed feelings burst forth.

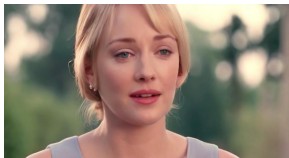

Daisy was also thrilled to see Gatsby. They reignited their past romance.

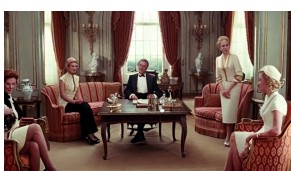

Gatsby made Daisy and her husband, Tom, into a room for a confrontation, hoping Daisy would leave with him.

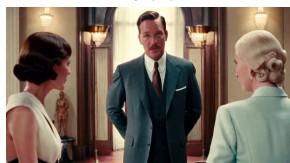

However, Tom had already investigated Gatsby's background. He began to publicly expose Gatsby.

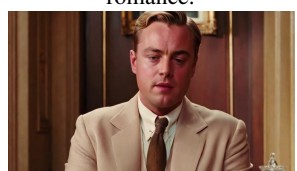

At first, Gatsby managed to remain calm.

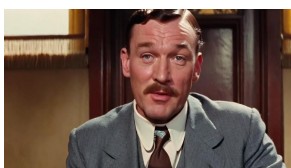

But Tom went on to reveal many unsavory facts about Gatsby.

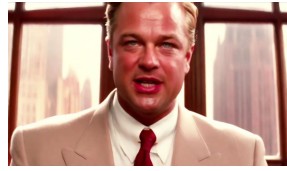

As a result, Gatsby completely flew into a rage.

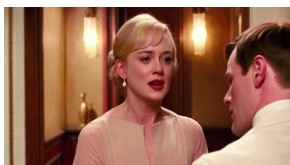

Witnessing this, Daisy also broke down in tears.

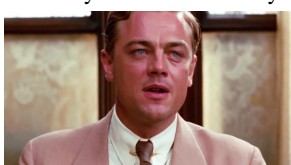

Gatsby still wanted Daisy to leave with him.

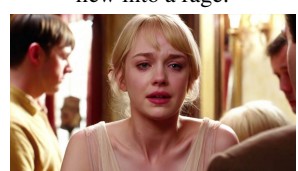

But Daisy was already in distress and ran out of the crowd.

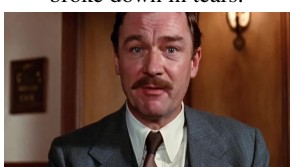

Tom stood there smiling, knowing he had won.

Figure 11: Additional Zero-Shot results.

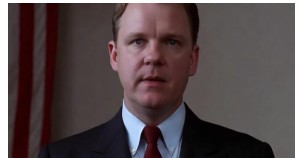

Andy was wrongfully convicted because he was suspected of murder.

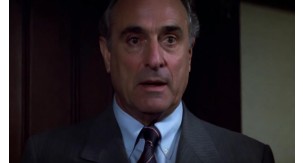

The lawyer didn't believe in Andy's innocence and sent him to prison.

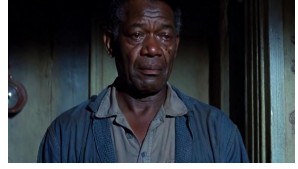

Meanwhile, Red was also undergoing a parole review.

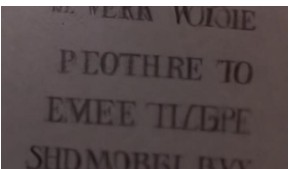

But unfortunately, his parole was denied.

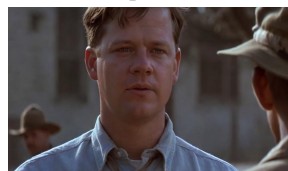

After entering prison, Andy and Red formed a connection.

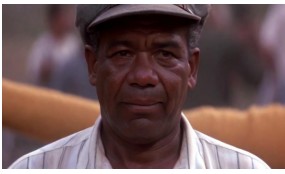

Red found Andy interesting and they quickly became friends.

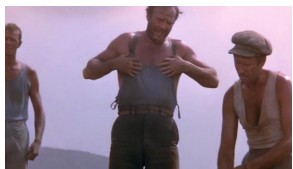

One day, Andy and his friends were working outside.

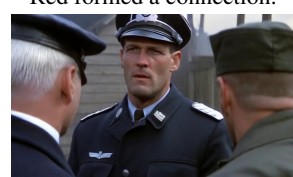

Andy heard the sheriff worrying about economic issues.

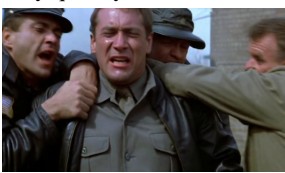

Ignoring the guard's objections, Andy seized the chance to offer help to the sheriff.

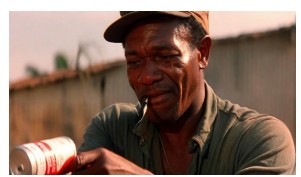

As a result, Andy won beer for his friends.

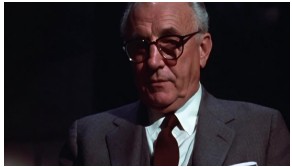

The warden heard of Andy and had him work for him.

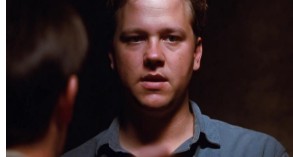

Andy agreed.

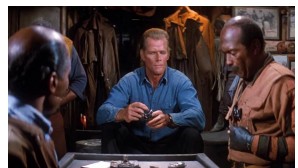

Andy took this opportunity to improve the quality of the library in the prison. Inmates' quality of life became better.

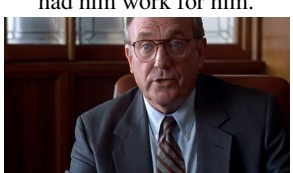

But the warden was no longer willing to let Andy leave prison because he knew too much about the warden's dirty work.

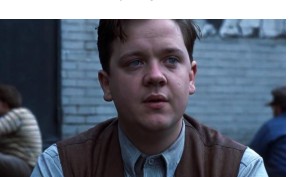

Andy told Red to look for something he left under a tree after his release.

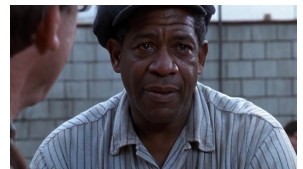

Initially, Red took this as a joke.

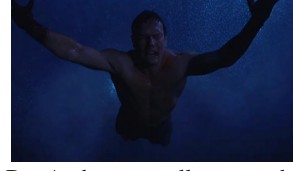

But Andy eventually succeeded in escaping from prison.

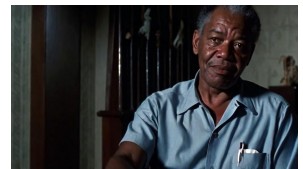

Soon after, Red's parole was also granted.

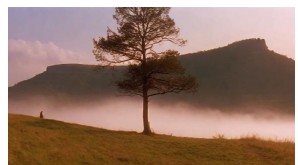

He found the oak tree that Andy had mentioned.

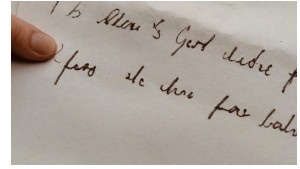

Beneath the tree, he discovered the letter Andy had left for him.

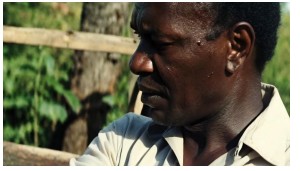

Red was happy because he knew they would meet again.

Figure 12: Additional Zero-Shot results.

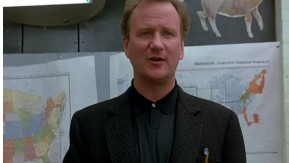

Gerald is a math professor, and one day he left a difficult problem on the blackboard.

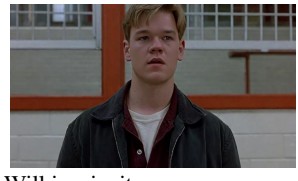

Will is a janitor on campus, and he noticed the problem the professor left behind.

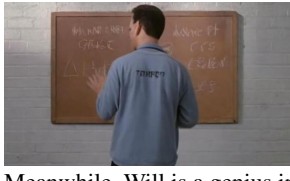

Meanwhile, Will is a genius in mathematics, and he quickly became immersed in thought.

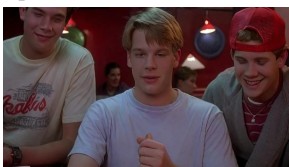

At night, Will would go to the bar to hang out with his friends.

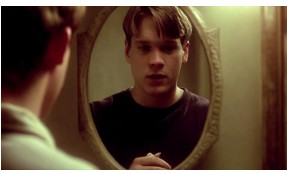

But after returning home, Will pondered the solution in front of the mirror.

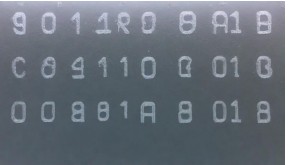

The next day, Will left the answer on the blackboard.

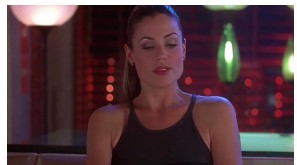

The professor was very surprised, but no students claimed they solved it themselves.

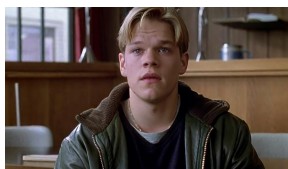

One day, a student troubled Will's friend at the bar.

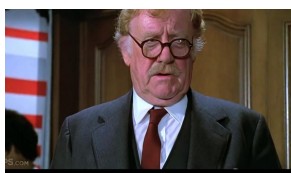

But Will managed to refute him with his talent.

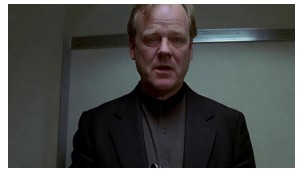

This helped him win Skylar's favor.

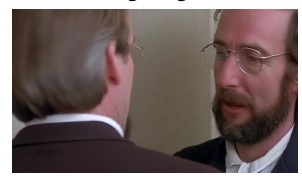

However, Will is a troubled youth. He was taken to court for fighting.

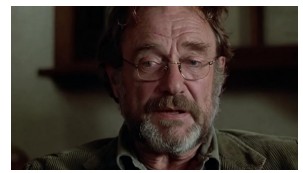

The judge intended to send him to prison.

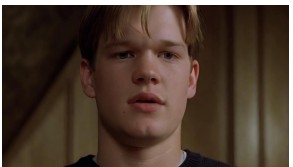

The professor was already aware of Will's situation and proposed that he study mathematics and undergo regular psychological evaluations so he wouldn't have to go to prison.

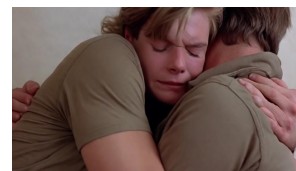

But many psychologists are insulted by Will. So Gerald had to turn to his old friend Sean for help.

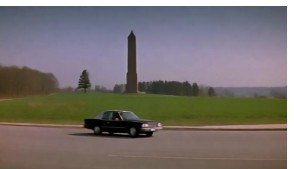

At first, Sean also found it difficult to cope with Will.

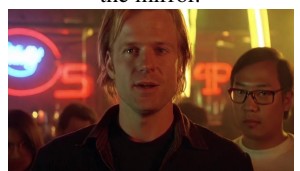

But he didn't give up. His sincerity eventually touched Will.

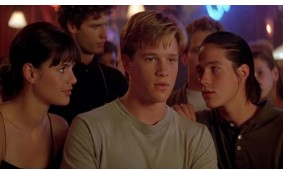

In the end, Will achieved redemption with Sean's support.

After that, Will embarked on a new journey in life.

Figure 13: Additional Zero-Shot results.

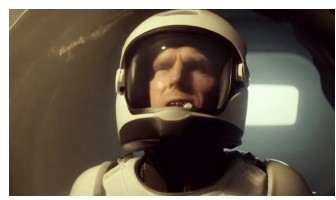 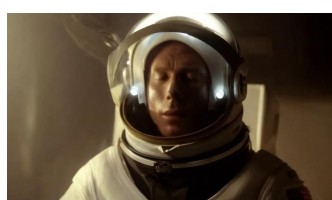 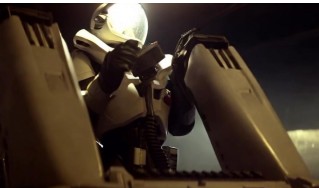

James is checking the spacecraft system.

This is his first mission. He is kind of nervous.

James launches the spaceship. They are about to land on a new planet.

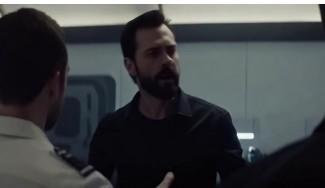 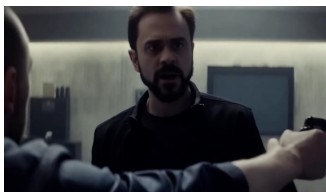 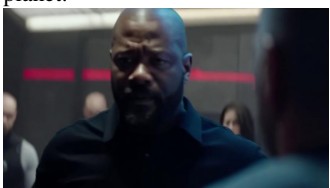

Meanwhile, the command center is debating whether they should land on this planet.

Steven believes that this planet is worth exploring.

However, others disagree with Steven's idea, as the planet is full of unknowns and could pose dangers.

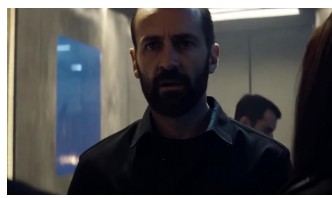 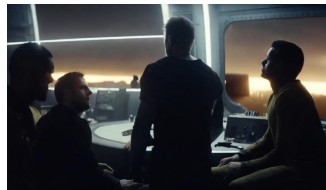 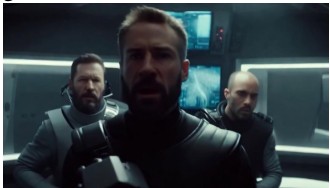

But Steven believes that high risks come with high rewards, and he stands by his opinion.

They have a heated debate.

In the end, Steven convinces everyone and sends the landing instructions to the spaceship.

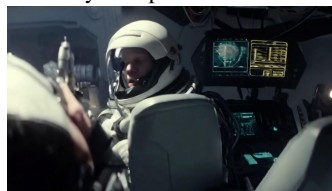 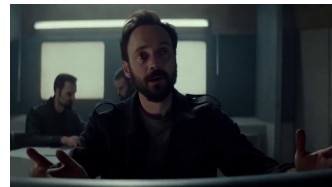 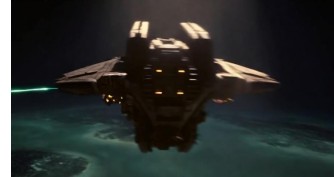

James and the astronauts accompanying him receive the instructions and begin preparing for landing.

Steven is directing James and the others from the command center as they prepare for landing.

The spaceship gradually approaches the planet's surface.

Figure 14: Additional Zero-Shot results.

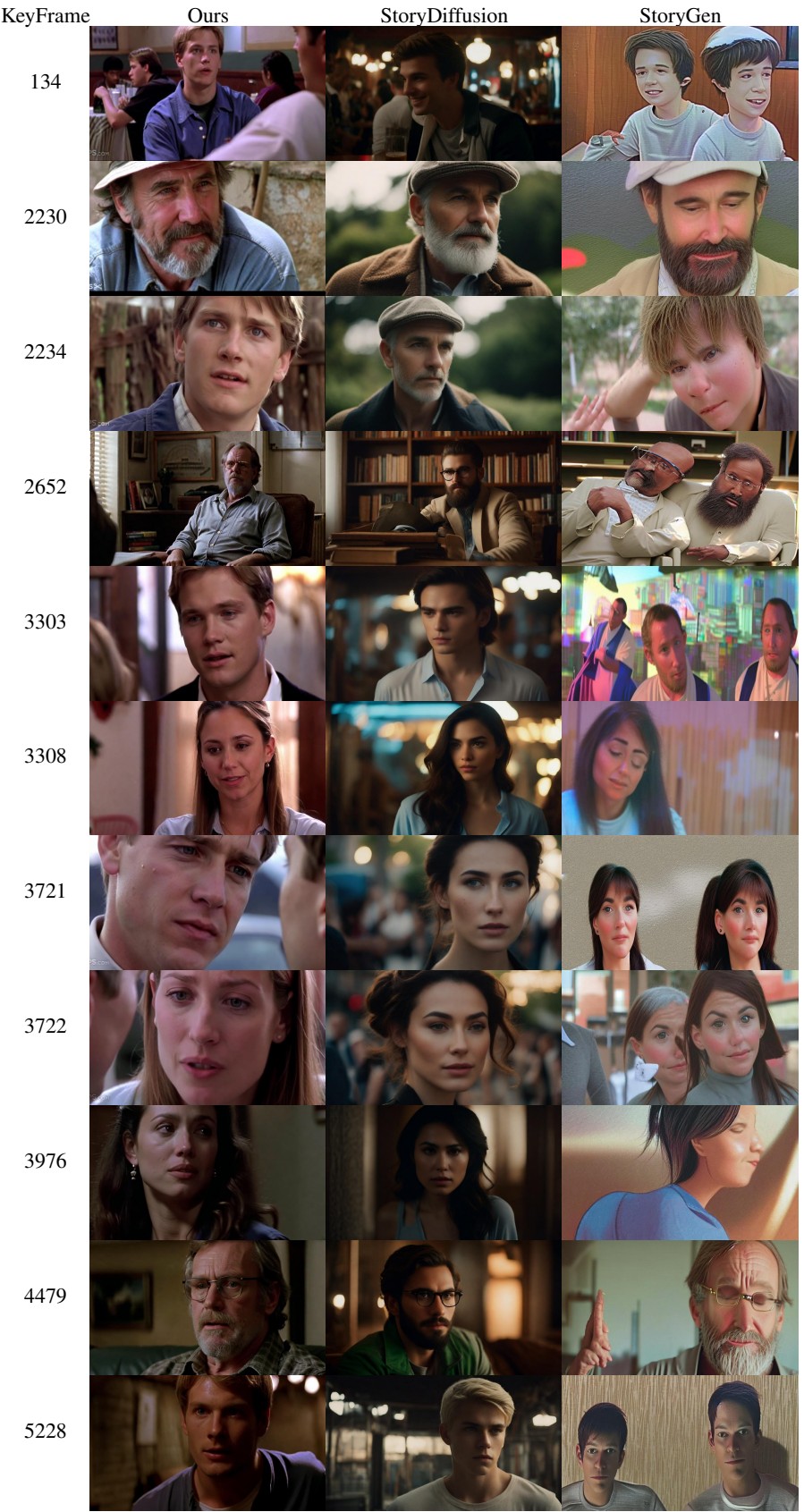

Figure 15: Additional story comparison results.

During training, we additionally introduce the $\ell_1$ and $\ell_2$ loss between the predicted image tokens and the ground truth tokens to improve the model's performance. However, we find that these losses are not on the same scale as the negative log-likelihood. The negative log-likelihood is way larger than the $\ell_1$ and $\ell_2$ loss, which reduces the influence of the $\ell_1$ and $\ell_2$ loss during the optimization. The difference in scale is because the constant term and covariance matrix of the negative log-likelihood are directly related to the dimension $d$. Concretely, for a variable $\mathbf{x}$ with mean $\mu$, the calculation of the negative log-likelihood is essentially represented as:

$$\text{NLL}(\mathbf{x}; \mu, \Sigma) = \frac{d}{2}\log(2\pi) + \frac{1}{2}\log|\Sigma| + \frac{1}{2}(\mathbf{x}-\mu)^\top \Sigma^{-1}(\mathbf{x}-\mu), \tag{8}$$

where $\frac{d}{2}\log(2\pi)$ is the constant term and $\Sigma$ is the covariance matrix. As $d$ increases, both the constant term and the size of the covariance matrix grow, resulting in a significant increase in the loss value. To bring the different losses to the same scale, we scale the negative log-likelihood loss by dividing it by the dimension $d$. Consequently, the negative log-likelihood loss in Section 3.4 is formulated as:

$$
\begin{aligned}
L_{nll} &= \mathbb{E}(-\log(p(\mathbf{e}|\mathbf{H}_{<t}, \mathbf{M}_t))) \\
&= \mathbb{E}\left(-\sum_{t=1}^{T}\sum_{i=1}^{L}\log p(e_{t,i}|e_{t,<i}, \mathbf{e}_{<t}, \mathbf{c}_{\leq t}, \mathbf{d}_{\leq t}, \mathbf{i}_{\leq t})\right) \\
&= \frac{1}{d}\cdot\mathbb{E}\left(-\log\left(\sum_{i=1}^{k}w_l^{(i)}\prod_{c=1}^{d}\mathcal{N}(e_{l,c}|m_{l,c}^{(i)}, v_{l,c}^{(i)})\right)\right).
\end{aligned} \tag{9}
$$

## B  EVALUATION METRICS

We propose new metrics to evaluate short-term (ST) consistency and long-term (LT) consistency. For short-term consistency, we select all the frames where a character appears. We then calculate the similarity of CLIP embedding of selected consecutive keyframes, as consecutive keyframes where the same character appears usually have high similarity. For long-term consistency, We calculate the proportion of generated frames where the character appears to the ground truth frames where the character appears. For example, if the GT frames where a character appears are 1, 2, 3, 4, 5, and the generated frames where the character appears are 1, 2, 3, 6, then the long-term consistency is calculated as 3/5. To select the frames where a character appears, we first select an image of the main character and use FARL to obtain the character's embedding. We then compute the similarity of this embedding with FARL embeddings of all other frames and select images above a certain threshold.

The Aesthetic Score (Schuhmann et al., 2022) is calculated based on the CLIP image features. Each image is first passed through the CLIP image encoder to obtain an embedding. This embedding is then fed into an additional small neural network, which predicts an aesthetic score.

## C  MULTI-MODAL SCRIPTS DATA PIPELINE DETAILS

**Multimodal script construction.**  In this chapter, we explain in detail how we construct the multimodal script data. Specifically, We collect a massive movie dataset, with movies coming from Condensed Movies (Bain et al., 2020) and others collected using the same methodology as Condensed Movies. The data consists of around 23,000 long videos, which subsequently results in 5 million keyframes with corresponding script annotations. The overall multimodal script synthesis is as follows:

- Extract keyframes from long videos at 1-second intervals, removing the beginning and end segments. These segments primarily contain sponsor messages, acknowledgments, and advertisements, lacking useful information. Subsequently, center-crop and resize each frame to a width of 896 pixels and a height of 512 pixels. Finally, leverage a Vision Language Model (Lu et al., 2024) to generate descriptions for the entire image.
- Using GroundingDINO (Ren et al., 2024a) with "Person" as the prompt, extract the top 5 characters from each frame. If the number of characters exceeds 5, the extra characters are ignored. If the number of characters is less than 5, there is no additional processing.

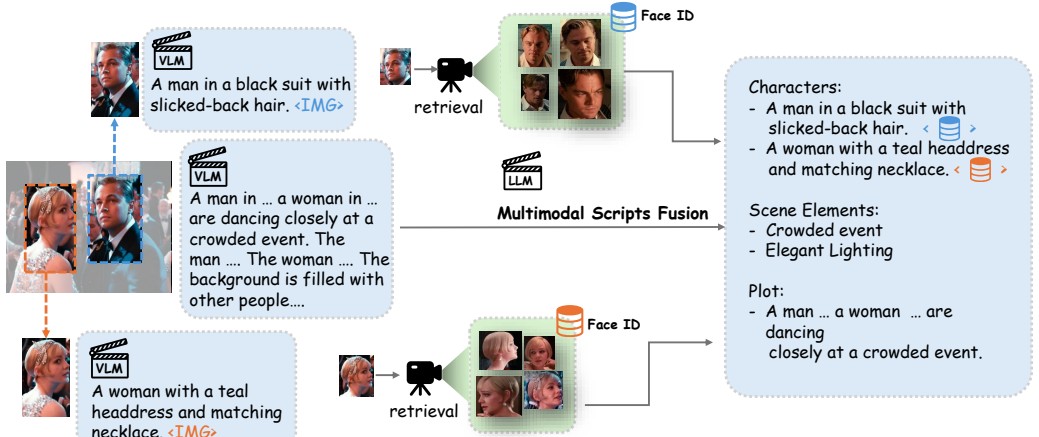

Figure 16: Multi-modal scripts data pipeline. We generate image captions using existing VLMs, and then further utilize LLMs to reformat the annotations into a structured script. We use the face retrieval method to extract different face embeddings of the same character, which benefits anti-overfitting data augmentation in Section 3.4.

- Employ an existing Vision Language Model (Lu et al., 2024) to obtain captions for each segmented character. The face of each character is extracted to construct the face ID bank.

- Use a Large Language Model, taking the captions for the entire image extracted in step 1 and captions for each character in step 3 as input, to derive Scene Elements and Plot, and ask the model to output in a multimodal script format. At this point, we have structured Scene Elements and Plot but still lack Character. We then format all the character captions corresponding to each frame into the Character section of the multimodal script.

- However, the character face IDs for each frame are derived from the same frame, which can lead to the model overfitting by directly copy-pasting characters into the generated results. To address this, we use FARL (Zheng et al., 2022) to randomly retrieve the same character's ID from the face ID bank as input. This strategy effectively mitigates the overfitting issue.

## D  ADDITIONAL RESULTS

In this section, we present additional generated stories and videos. Our method exhibits strong generalization capabilities, producing high-quality results with inputs **not included in the training set**. Our method can understand text and character inputs, capable of generating results that align with the storyline while ensuring character identity consistency. The overall scene consistency is also well-preserved, as our autoregressive model is able to leverage context information as the reference. Most importantly, we are the **first** method to generate extremely long contents while preserving both short-term and long-term consistency. Besides, we even ensure that the IDs of different characters remain consistent before and after a camera change.

**Generation process.**  Our autoregressive model is similar to a multi-turn dialogue model. Specifically, we input the multimodal script corresponding to a keyframe, and the model autoregressively outputs two compressed tokens. Then, based on this, we input the multimodal script for the next frame to obtain the next two compressed tokens, without losing historical information. After obtaining all the compressed tokens, we input them along with the face embeddings and description embeddings from the multimodal script into the decoder to generate the keyframe. The keyframes are subsequently leveraged by image-to-video model to generate videos.

**Additional results.**  Our method effectively preserves both short-term and long-term consistency for multiple characters. We present numerous results in Figure 10, 11, 12, 13 and 14. Our method is able to preserve multiple characters within long contents without the need for any fine-tuning.

**Step1: Frame Caption**

**Step1 User input:**

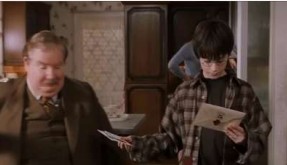

Provide a detailed yet concise description of the image that allows DALL-E or Midjourney to recreate it, focusing on key elements such as subjects, background, and objects. Describe colors, textures, lighting, and spatial relationships between objects, ensuring accuracy. If there is motion, vividly depict its direction and nature. Aim for a clear, comprehensive description, but avoid unnecessary length.

**Step1 VLM Output:**

In a warmly lit room, a man in a brown suit and a young boy in a plaid shirt and glasses stand facing each other. The man holds a pen, poised to write on a piece of paper, while the boy examines a letter with a crest on it. Behind them, a woman in a blue shirt and glasses leans against a wooden cabinet, observing the scene.

**Step2: ID Caption**

**Step2 User Input:**

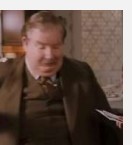 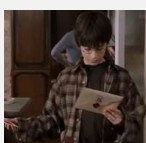

Provide a detailed yet concise description of the character that allows DALL-E or Midjourney to recreate it, focusing on key elements such as subjects, and objects. Describe colors, textures, lighting, and spatial relationships between objects, ensuring accuracy. If there is motion, vividly depict its direction and nature. Aim for a clear, comprehensive description, but avoid unnecessary length. Better restrict in 100 words

**Step2 VLM Output (List):**

An image of a man in a brown suit and tie, appearing surprised, in an indoor setting with a blurry background. 

A young man in a plaid shirt reads a letter with an owl stamp, in a room with wooden cabinets. 

**Step3: LLM Fusion**

**Step3 User Input:**

Transform the provided description {**Step1 VLM Output**} and its characters{**Step1 VLM Output (List)**} into a structured format by identifying Scene Elements, and Plot. Ensure to replace placeholders within brackets `[ ]` with specific details from the text. Character descriptions should be simple, without detailed elaboration. The plot summary should succinctly encapsulate the central event or emotion in no more than 50 words. Follow the structure below for formatting your output:
  Scene Elements:
  - [Description of First key element of the scene]
  - [Description of Second key element of the scene]
  - [Description of Third key element of the scene]
  ...

  Plot:
    - [A concise summary focusing on the main character event or emotion, within 60 words].

**Step3 LLM Output:**

Scene Elements:
- Warmly lit room
- Man holding a pen
- A letter with a crest

Plot:
- In a warmly lit room, a man and a young boy stand facing each other. The man holds a pen, poised to write on a piece of paper, while the boy examines a letter with a crest on it. Behind them, a woman observing the scene.

**Step4: Compose**

**Step4 Output:**

Characters:
- An image of a man in a brown suit and tie, appearing surprised, in an indoor setting with a blurry background. 
- A young man in a plaid shirt reads a letter with an owl stamp, in a room with wooden cabinets. 

Scene Elements:
- Warmly lit room
- Man holding a pen
- Aletter with a crest

Plot:
- In a warmly lit room, a man and a young boy stand facing each other. The man holds a pen, poised to write on a piece of paper, while the boy examines a letter with a crest on it. Behind them, a woman observing the scene.

Figure 17: The overall data pipeline and corresponding prompts we use.

Even when the scene changes, our method effectively maintains the same character ID even during camera switches. Due to space limitations, we only select certain excerpts to showcase here. Please refer to the TITANIC generated by our MovieDreamer in supplementary material for longer results.

**ID preservation.** We showcase our ability to preserve character ID in Figure 18. Our method exhibits strong generalization capabilities, allowing it to generate target characters without any further training.

**Video results.** The generated keyframes serve as the condition to generate videos. We show the results generated with our improved video generation model in Figure 19. Our keyframes have

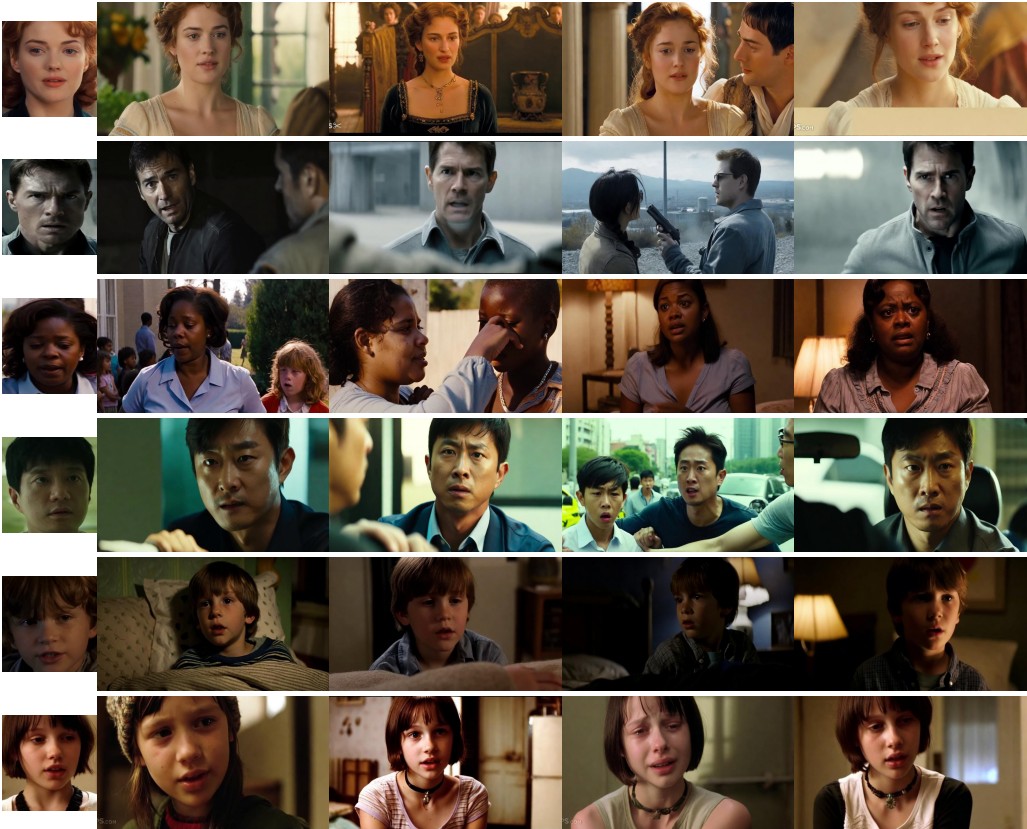

Figure 18: ID Preservation results. Our method demonstrates the great ability to preserve ID. The compressed token for each keyframe is generated with the corresponding multimodal script as input, and the keyframes are rendered with compressed tokens, face embedding and description embedding as input. We show the target ID on the left.

successfully preserved both short-term and long-term temporal consistency, facilitating the seamless rendering of these frames into a cohesive and harmonious video.

*Again, we would like to highlight that our primary contribution lies in the **hierarchical** controllable very long generation of visual contents using MLLM for keyframe generation and diffusion model for video rendering.* Any high-quality image-to-video model can be integrated with our approach. We showcase video results generated using higher-quality image-to-video models with our generated keyframes as input in Figure 20, which strongly demonstrates the effectiveness and flexibility of our method. We believe our approach is highly promising has immense potential.

**Different shot types and styles.** We find that although we did not explicitly model the camera shot types, the model can implicitly learn various shot types based on the multimodal script. The model is capable of selecting the most appropriate shot type based on the input. The shot types of anchor frames further assist the image-to-video model in generating videos with various shot types. Our method is also capable of generating cartoon results. We showcase the results in Figure 21.

# E ADDITIONAL EXPERIMENTS

## E.1 ADDITIONAL COMPARISONS

We showcase additional comparisons in Fig.15. Our method effectively preserves both short-term and long-term consistency, maintaining the overall style and character identity. Other methods exhibit increasing inconsistencies as the number of frames grows, such as changes in character ap-

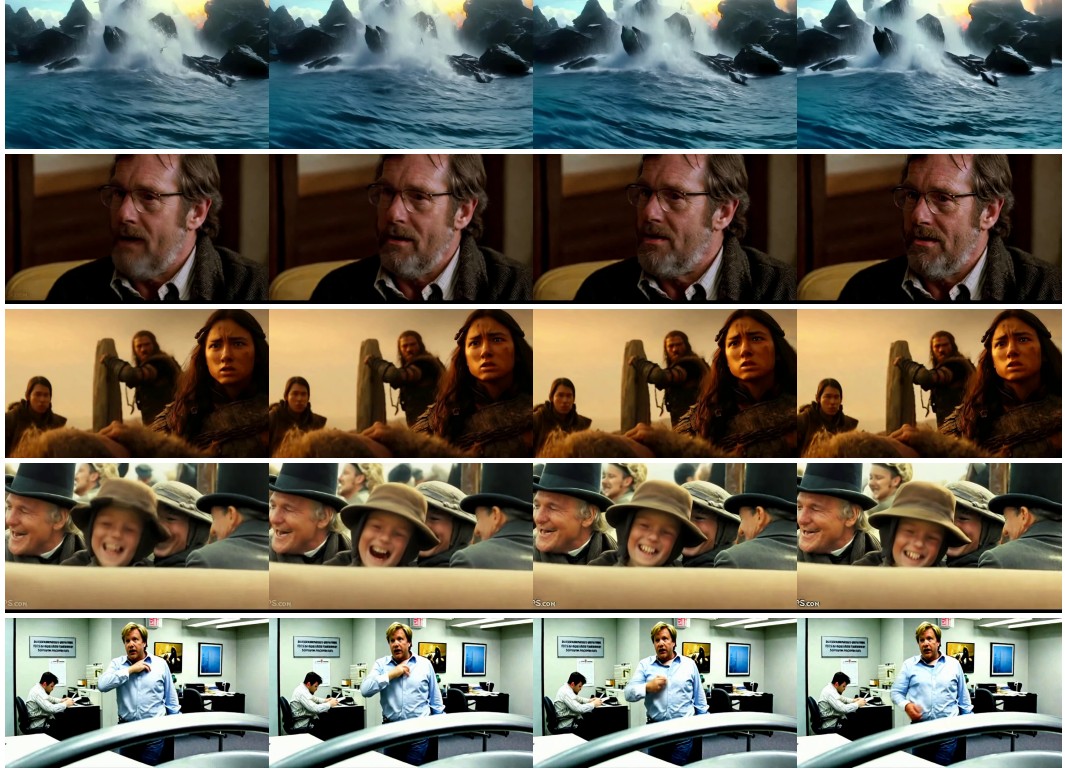

Figure 19: Example video results with our video generation method.

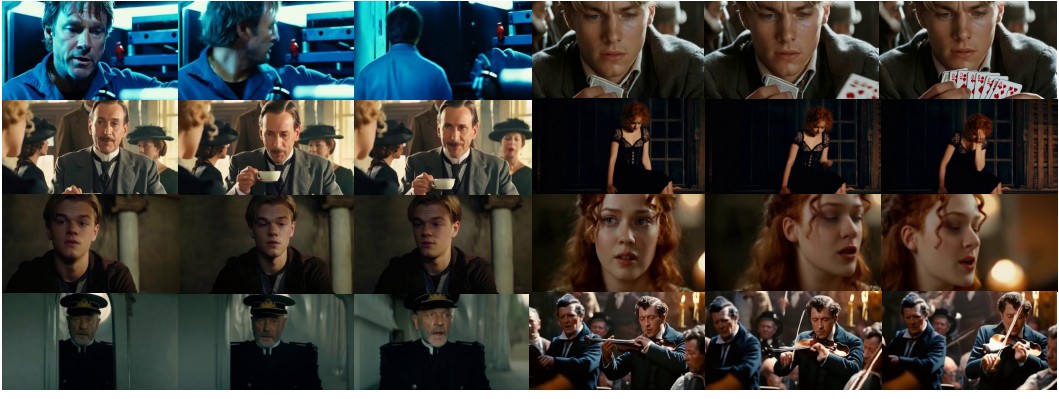

Figure 20: Example video results using existing better image-to-video model LUMA. Existing image-to-video models can be integrated with our method to produce high-quality ultra-long videos, which demonstrates the flexibility and effectiveness of our MovieDreamer.

pearance and shifts in style. Our method can generate extremely long content while nearly perfectly maintaining the consistency of multiple characters throughout.

## E.2 ADDITIONAL ABLATIONS ON THE NUMBER OF COMPRESSED TOKENS

**Our goal and an important tradeoff.** While more tokens may yield better results, they sacrifice the context length of autoregression. Since we aim for long video generation, using as few tokens as possible is a reasonable design choice. Our goal is to achieve the highest quality images with a minimal number of tokens.

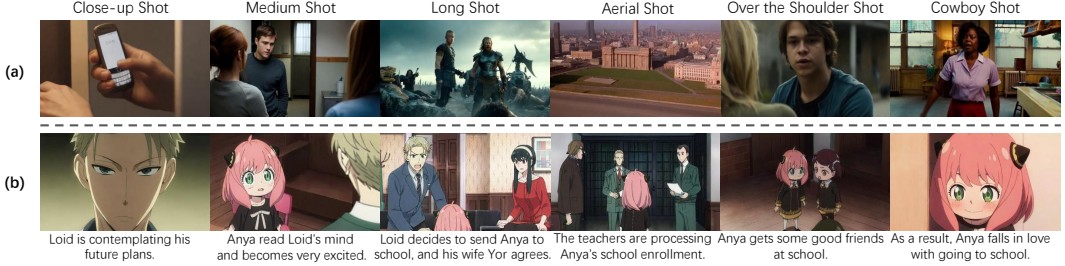

Figure 21: **(a)** Although our method does not explicitly model shot types, it can still implicitly learn various shot types and generate high-quality results with suitable shot types, demonstrating the potential of our approach. **(b)** Our method can also generate cartoon results and interleaved characters.

Table 3: Quantitative ablation study on the number of compressed tokens.

| Method | CLIP-sim↑ | Inception↑ | Aesthetic↑ | FID↓ |
|---|---|---|---|---|
| 2 tokens | 0.825 | 8.235 | 6.378 | 0.406 |
| 8 tokens | 0.841 | 8.193 | 6.457 | 0.348 |

**Why do we choose 2 tokens.** Existing work Paint-by-Example (Yang et al., 2022) demonstrates that a single token can effectively reconstruct the original image. Building on this discovery, we added a few additional tokens. We compared the reconstructed images using 8 tokens and 2 tokens, with the metrics shown in the accompanying table below. The metrics indicate that increasing the number of tokens does not offer significant improvements in results. As shown in Table 3, we calculate the inception score, aesthetic score as well as FID score of images of different numbers of tokens. We further compare the CLIP similarity between reconstructed images and target images. The evaluation metrics demonstrate that the results of 8 tokens are slightly improved. But considering our objective of long content generation, this slight improvement is not enough for us to prioritize it when making trade-offs regarding sequence length and our limited computational resources, as 8 tokens will lead to a much shorter sequence length and demand more computational overhead. Consequently, we eventually chose 2 tokens. But we believe that if there are enough computational resources, 8 tokens will be a better choice.

## E.3 ADDITIONAL ABLATIONS ON CONTINUOUS TOKEN SUPERVISION

We conduct additional experiments to further explore the importance of continuous token supervision as well as the L1 and L2 losses. The evaluation metrics are shown in Table 4. As visualized in Figure 22, when the model is trained using only continuous token supervision as the objective function, it tends to produce poor results with characters placed at the edges of the image. However, this problem is effectively resolved when L1 and L2 are introduced as additional objective functions during training. Our experiments demonstrate that both continuous token supervision and the L1 and L2 losses play crucial roles in optimizing the model.

Table 4: Quantitative comparisons of results trained with different losses.

| Method | CLIP ↑ | Inception↑ | Aesthetic↑ | FID↓ | ST↑ | LT↑ |
|---|---|---|---|---|---|---|
| Con-L1-L2 | 19.584 | 9.698 | 6.093 | 2.043 | 0.646 | 0.814 |
| Con-L1-L2-ref | **20.071** | **9.842** | **6.288** | **1.912** | **0.701** | **0.893** |
| L1-L2 | 18.833 | 8.609 | 5.893 | 2.714 | 0.606 | 0.755 |
| L1-L2-ref | 19.431 | 8.774 | 5.979 | 2.560 | 0.616 | 0.776 |
| Con | 16.882 | 7.463 | 5.352 | 3.283 | 0.576 | 0.701 |
| Con-ref | 17.039 | 7.580 | 5.411 | 3.136 | 0.571 | 0.708 |

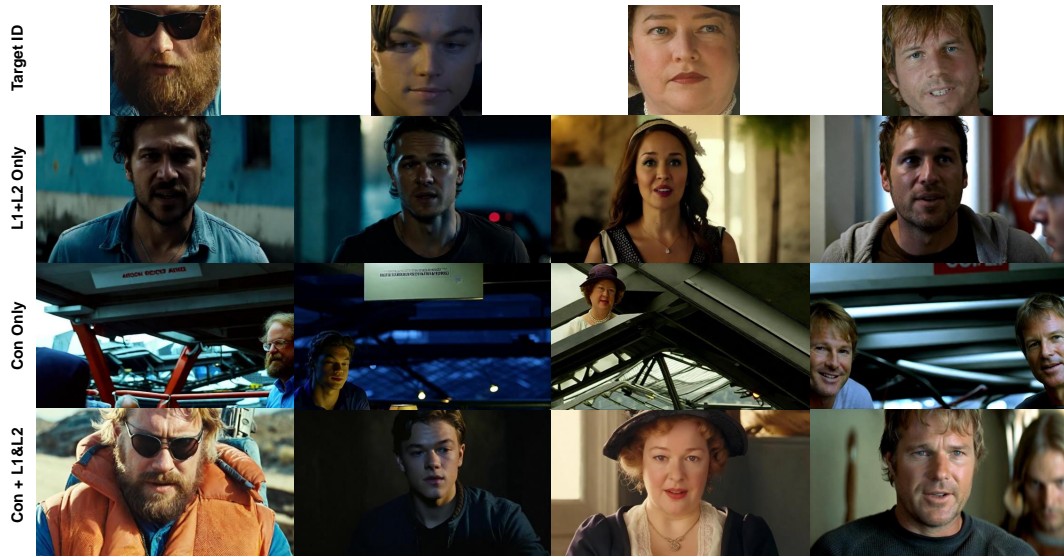

Figure 22: Comparison between using only continuous token supervision and incorporating both L1 and L2. The results show that relying solely on continuous token supervision leads the model to place characters at the edges of the image, resulting in low-quality keyframes. With L1 and L2 only, the model is able to learn layout information but struggle to maintain target ID. With continuous token supervision, L1 and L2, the model is able to generate results with reasonable layout and preserve the target ID effectively.

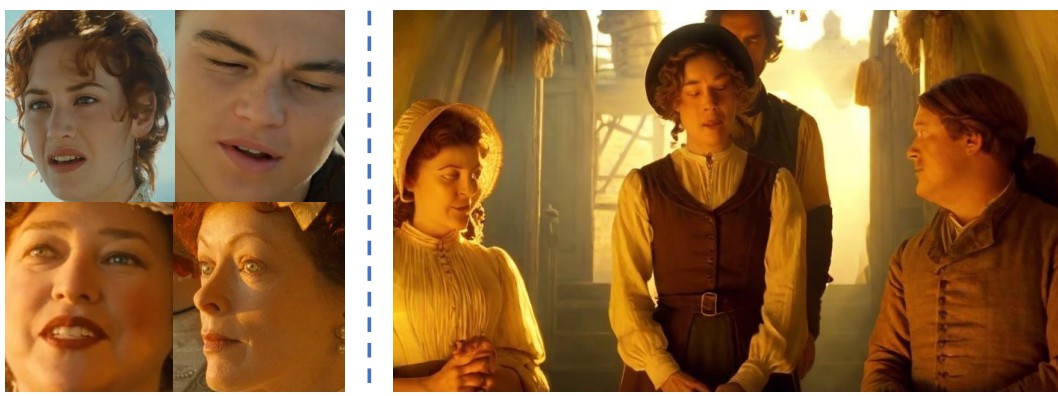

Figure 23: When there are too many characters in one frame, the model sometimes confuses the character IDs.

## F    LIMITATIONS AND FUTURE WORKS

Despite our method demonstrating strong potential in generating long sequences, it still has some limitations. (1) We employ LongCLIP to compress each sentence of descriptive text into a single token. While this strategy effectively reduces the sequence length, it provides only a coarse representation of each sentence, which causes information loss. (2) Our method's ability to preserve object consistency is limited. We did not construct the multi-modal script with scene information, as the costs far exceed our capacity. Moreover, as shown in Fig. 23, if there are too many people in one frame, our method is prone to mixing the appearance of characters. (3) Analogous to other large models, our approach requires substantial data and computational power for training. (4) Currently, we generate 128 keyframes at a time and run multiple iterations of this process to create long content. However, our goal is to extend the maximum token length, fundamentally addressing the

long sequence generation by using long sequence generation methods in LLMs. Due to resource constraints, we are unable to pursue this approach at present and leave it in future work.

**Social impact.** Our method can generate high-quality long stories and videos, significantly lowering the barrier for individuals to create desired visually appealing content. However, it is essential to address the negative potential social impact of our method. Malicious users may exploit its capabilities to generate inappropriate or harmful content. Consequently, it is imperative to emphasize the responsible and ethical utilization of our method, under the collective supervision and governance of society as a whole. Ensuring appropriate usage practices requires a collaborative effort from various people, including researchers, policymakers, and the wider community. Furthermore, the code, model, as well as the data, will be fully released to improve the development of related fields.

**Acknowledgements** This work was supported by the National Key R&D Program of China (NO.2022ZD0160101) and the National Natural Science Foundation of China (No. 62206244).

