# OpenReview forum: "MovieDreamer: Hierarchical Generation for Coherent Long Visual Sequences"
_ICLR.cc/2025/Conference — ICLR 2025 Poster_

### Official Review · Reviewer_Jjeo · 2024-11-02

**Soundness:** 3
**Presentation:** 3
**Contribution:** 3
**Rating:** 6
**Confidence:** 5

**Summary:**

The paper proposed a novel hierarchical video generation framework MovieDreamer, which is able to generate long-duration videos with high visual fidelity. The framework consists of a SDXL-based visual autoencoder and a multimodality VLM. The autoencoder focuses on obtaining the visual tokens of input images, as well as rendering targeted keyframes, while VLM autoregressively predicts visual tokens. Quantitative evaluation shows that the proposed method outperforms sota in long video generation.

**Strengths:**

1. The proposed framework propose a hierarchical way to combine autoregressive modeling and diffusion image generator together for long-term complex video generation.
2. The paper proposed several novel techniques, e.g., continuous token supervision, anti-overfitting strategies, id-preserving, to improve the performance of the framework.
3. The proposed method achieved state-of-the-art performance on quantitative evaluation.

**Weaknesses:**

1. L212: Eq1, the DAE has not been clearly defined in the paper. Also, in Fig2, I could not find DAE in the pipeline. I hope the authors could clarify this.
2. It is a bit confusing whether Eq1 is the real training loss or not? According to Sec3.3 and Fig2, the input of SDXL should be image token, face embedding and description embedding, which does not align with Eq1. It is better to clarify the final training loss.
3. It is not clear how random masking strategies work during training and inference stages.
4. The high dropout rate (50%) seems to be unusual. Are all the parameters updated in VLM? Have the authors tried only tuning part of the network?
5. L323: if the feature of the first frame is always used, will generated character be limited to small motion, and limited content? For example, for large rotation and movement, the performance of the proposed method may drop?
6. What are the frame numbers of full-length video?
7. From Fig8, while compared to zero-shot setting, the few-shot setting indeed improves the ID-preserving ability. However, there is still an obvious gap between target ID and generated image. What could be the potential reason for this? Any solution?
8. What is the training cost? And what are the inference time and GPU memory requirements?
9. For quantitative evaluation, it seems that the reported metrics all focus on image level. They can not evaluate how good the performance of spatio-temporal modeling. It is better the authors could evaluate the proposed method using other metrics for video generation.

**Questions:**

See weaknesses

---

> ### Author Response · Authors · 2024-11-13
> **Official Comment by Authors (1/2)**
>
> Thank you for your valuable feedback and suggestions. To clarify our approach, we first provide a brief summary of our video generation process:
> 1. The autoregressive model predicts keyframe tokens.
> 2. The DAE decoder decodes the keyframe tokens into keyframe images.
> 3. Leveraging existing image-to-video methods to generate video clips for each keyframe, resulting in a long video.
>
> **W1&W2. Questrions about DAE.**
>
> Thank you for pointing out the confusion. DAE refers to the Image Renderer Training in Fig. 2. Our DAE training is conducted in three stages:
> 1. In the first stage, we train on natural images (768x768) using Eq. 1 as the loss function.
> 2. In the second stage, we train on movie images (896x512) using Eq. 1 as the loss function.
> 3. In the third stage, we further incorporate face embedding and description embedding and train the model on movie images (896x512). The training loss for this stage becomes :
> $
> \mathcal{L}\_{DAE} = \mathbb{E}\_{\mathbf{x\_0}, \epsilon \sim \mathcal{N}(0, \mathbf{I})} \left\| \epsilon - \mathcal{D} \left( \mathbf{z\_t}, t, \mathcal{E}(\mathbf{x\_0}), \mathbf{f}, \mathbf{d} \right) \right\|\_2^2,
> $ where $\mathbf{f}$ and $\mathbf{d}$ is the face embedding and description embedding, respectively.
>
> We apologize for not clarifying this point. We will refine the figure and the explanation in the revised paper. Further details can be found in Appendix A.
>
> **W3. Random masking.**
>
> The random masking strategy is used only during training. It is applied in both the DAE and autoregressive model training. In the third stage of DAE training, we input concatenated image tokens, face embeddings, and description embeddings. With a probability of 0.15, we randomly replace each token with a zero token. During autoregressive model training, we randomly prevent the model from attending to masked tokens with a probability of 0.15 by applying attention mask. This strategy helps the model learn to infer missing content from the available information, improving its performance and reducing overfitting. During inference, random masking is not used.
>
> **W4. High dropout rate.**
>
> All parameters in the VLM are updated. We didn't try only tuning part of the model.  We conducted detailed experiments and found that a high dropout rate is indeed beneficial in our setting.
>
> | Validation Loss/step  | 2500 | 5000 | 7500 | 10000 | 12500 | 15000 |
> |:-------------:|:------------:|:------------:|:------------:|:------------:|:------------:|:------------:|
> | Dropout | 1.142  | 1.018 | 0.959 | 0.937 | 0.921 | 0.898 |
> | No Dropout | 1.187 | 1.084 | 1.036 | 0.976 | 0.982 | 0.973 |
>
> **W5. Questions about image-to-video model.**
>
> - **i2v (image-to-video) is not our primary focus; the emphasis is on ultra-long generation. i2v is just a tool we use.** The core of i2v lies in generating high-quality short videos from images, whereas the core of our work is to generate coherent and consistent keyframes, and then use i2v to produce long videos. We choose to use SVD as our i2v method by default, as it is widely adopted in many existing works. However, it tends to produce results with limited motion. Therefore, this is an issue with SVD itself, rather than a limitation of our approach.
>
> - Our main contribution is in exploring ultra-long video generation. We have emphasized in the paper that our method can be combined with any image-to-video model, and we demonstrate in the supplementary video that using other, more advanced i2v models can produce higher-quality, action-rich results. Therefore, we respectfully believe that the performance of our method would not be affected by any particular i2v approach. On the contrary, the performance of our method is determined by **the upper bound of the entire i2v field.**
>
> **W6. The frame numbers of full-length video.**
>
> - Currently, our model supports generating up to 128 keyframes at once. The total number of frames in the full-length video depends on the specific image-to-video method used. When employing the SVD-based approach outlined in the paper, the maximum number of video frames is 6400.
>
> - However, if the storylines can be seamlessly connected between each long video, the long videos we generate can also be combined to form even longer coherent videos. Therefore, to some extent, the number of frames in our full-length video could be unlimited.

---

> ### Author Response · Authors · 2024-11-13
> **Official Comment by Authors (2/2)**
>
> **W7. Gaps that sometimes occur and solution.**
>
> - First, the target ID in Fig. 8 is not meant to generate an exact match with the target ID, but rather to ensure that the generated character can be recognized as the same person as the target ID. We believe we have successfully achieved this goal.
>
> - In our opinion, the gap that sometimes occurs is due to our data not reaching the model's capacity. We are confident that scaling up the model with more data will address this issue.
>
> **W8. Training and inference cost.**
>
> - Training cost has been discussed in Appendix A. For DAE, it takes 3 weeks with 6 NVIDIA A800 GPUs. For Autoregressive model, it takes 3 days with 4 NVIDIA H100 GPUs.
> - For inference, both the 24GB and 80GB GPUs are capable of running the model. With the 80GB GPU, generating 128 keyframes takes approximately 3 minutes. This is because, during inference, all keyframe tokens are first generated, and then they are batched together for rendering into images. On the 24GB GPU, inference takes about 6 to 7 minutes, as it requires some off-the-shelf memory-efficient libraries, and during image rendering, smaller batch sizes are needed due to memory constraints.
>
> **W9. Metrics on spatial-temporal quality.**
>
> We have considered using metrics such as KVD, FVD to evaluate the spatio-temporal quality of videos. However, as discussed earlier regarding I2V in W5, we believe that video-related metrics primarily assess the image-to-video model's ability to generate short videos, rather than evaluating the core contribution of our work: ultra-long video generation. By taking both Table 1 and Table 2 into account, we believe that they already provide a comprehensive evaluation on long video level. Therefore, we respectfully believe that spatio-temporal metrics do not provide significant help in evaluating our core contribution.
>
>
> We sincerely thank you for your time and thoughtful feedback. We are happy to provide any further clarifications or engage in additional discussions to address any remaining concerns!

---

> > ### Comment · Reviewer_Jjeo · 2024-11-24
> > **Official Comment by Reviewer Jjeo**
> >
> > I thank authors for the rapid reply. I have fully read the response. The authors have addressed most of my concerns, but there are still some unsolved issues.
> >
> > 1. I think the ID-perserving problem has not been fully addressed. According to the authors, to solve this problem, it only requires larger-scale model and more training data. Since CogVideoX-5B is an open-sourced 5B model which also supports I2V, I was wondering wether the auhtors have tried to replace the original SVD with it to improve the performance?
> >
> > 2. The authors think "spatio-temporal metrics do not provide significant help in evaluating our core contribution", which I could not agree. As a video generation work, it is important to evaluate the consistency in the generated results, especially for long video generation. Except KVD and FVD, we still have some evaluation benchmarks such as VBench, which is designed specifically for T2V and I2V. I was wondering whether such metric fits the context here.

---

> ### Author Response · Authors · 2024-11-24
> **Address the Remaining Concerns**
>
> We sincerely appreciate your concerns and are delighted to have the opportunity to engage in this discussion!
>
> ---
> **1. Regarding Fig. 8 and the associated issues**
>
> We believe there might be some misunderstanding here.
> - **The results in Fig. 8 are the [predicted keyframes] at intervals of a few minutes from the autoregressive model, rather than a few seconds of continuous video generation.**
> - From our experiments, we found that the data we collected has not fully reached the capacity of the autoregressive model. Therefore, we believe scaling up the autoregressive model can address this issue. **This problem is unrelated to the i2v model.**
>
> **Of course, we have tried using different i2v models to generate results (including CogVideoX).** We compared the video results generated by various methods and ultimately chose to showcase LUMA (see supplementary materials) because it produces the best results. Replacing the original SVD with LUMA leads to significant improvements. Similarly, replacing SVD with CogVideoX-5B also improves the results. This highlights the advantage we have repeatedly emphasized: the quality of our videos depends on the state-of-the-art methods in the i2v field.
>
> However, the problem you mentioned in Fig. 8 stems from the limitations of the autoregressive model used for generating keyframes, which still has potential for improvement because the data we collected does not fully reach the model's capacity. It is independent of the i2v methods. Thus, using different i2v methods cannot solve this problem. And we are already focused on gathering more refined data to enhance the performance.
>
> ---
> **2. Regarding the evaluation with spatial-temporal metrics**
>
> We understand your concern, but we believe there may be some misunderstanding here as well.
>
> As we have repeatedly emphasized, our method can be combined with any existing i2v methods. Therefore, when evaluated with spatial-temporal metrics, it is actually the i2v methods being assessed. For instance:
> - If we use SVD, then VBench evaluates SVD.
> - If we use CogVideoX-5B, then VBench evaluates CogVideoX-5B.
> - If we use Gen2 or PIKA, then VBench evaluates Gen2 or PIKA.
>
> This is why we believe these spatial-temporal metrics are more focused on evaluating the i2v methods themselves, rather than our key contributions.
>
> **As for consistency, we have already provided evaluations of short-term and long-term consistency in Table 1 and explained in detail how these metrics are computed in Appendix B.** We believe this adequately evaluates consistency.
>
> > Finally, if you still believe that evaluating our method using metrics like VBench is necessary, we are more than happy to make every effort to provide such results. :)
>
> ---
> Thank you again for your feedback! We consider this discussion very enjoyable and valuable! Please do not hesitate to share further concerns, and we would be very delighted to address them!

---

> ### Author Response · Authors · 2024-11-25
> **Vbench Evaluation**
>
> Dear Reviewer,
>
> As soon as we received your previous comment, we immediately began using VBench to evaluate our method.
>
> Although our perspectives may differ, we deeply respect your emphasis on the importance of metrics like those in VBench. We have made every effort to further evaluate our method using the i2v-related metrics from VBench. :)
>
> |   Method   |   Video-Image Subject Consistency   |   Video-Image Background Consistency   |   Subject Consistency   |   Background Consistency   |   Motion Smoothness   |   Dynamic Degree   |   Aesthetic Quality   |   Imaging Quality   |   Temporal Flickering   |
> |:------------:|:------------:|:------------:|:------------:|:------------:|:------------:|:------------:|:------------:|:------------:|:-------------:|
> |    Ours     |    98.27%     |    98.42%    |    97.35%     |    97.92%     |    99.32%     |    73.81%     |    60.52%    |    67.92%     |     98.01%     |
> |    StreamingT2V     |    96.58%    |    96.12%     |    93.93%     |    95.51%     |    97.25%     |    17.18%    |    41.16%    |    62.34%     |     96.38%     |
> |    SEINE     |    97.17%    |    97.55%    |    94.98%     |    96.59%     |    98.41%    |    49.60%     |    58.25%     |    63.31%    |     96.47%     |
>
> If you have any additional concerns, please do not hesitate to let us know! :)

---

> > ### Comment · Reviewer_Jjeo · 2024-11-25
> > **Official Comment by Reviewer Jjeo**
> >
> > I thank authors for the quick response and further clarify the misunderstanding. My concerns have been addressed. I am happy to accept the paper.

---

### Official Review · Reviewer_4ibH · 2024-11-03

**Soundness:** 3
**Presentation:** 3
**Contribution:** 3
**Rating:** 8
**Confidence:** 4

**Summary:**

Existing diffusion model-based video generation models can only support short video generation and lack understanding of narrative structures and plot progressions. Therefore, this paper explores long-story video generation by leveraging the inference capability of autoregressive generation and video diffusion models’ rendering capability. Specifically, this paper proposes a hierarchical framework and a structured multimodal story input representation. The framework fine-tunes the multimodal large model LLAVA to generate compressed tokens of keyframes and fine-tunes SDXL as a decoder to generate keyframe images from the compressed tokens. Finally, SVD is used to generate the video. Additionally, the paper proposes a few-shot training method, enabling the model to achieve customized generation. Extensive experiments compare the proposed method with existing story keyframe generation methods, demonstrating superior performance.

**Strengths:**

1.	The proposed framework seems promising for generating long video.
2.	The proposed method can effectively generate character-consistent story keyframes, and the experimental results show the superiority of the proposed method to other state-of-the-art methods.

**Weaknesses:**

The overall writing of the paper is relatively clear, and the experimental results are promising. However, some of the descriptions appear to be incorrect or misleading, and the experiments are also not entirely comprehensive.

1. In line 242, why are the parameters of the GMM stated as 2kd means and 2kd variances? According to the reference work, each GMM parameter should contain kd means, kd variances, and k coefficients for each compressed token.

2. What would the results look like if Eq. 2 were used as the sole objective function? There doesn't appear to be a corresponding analysis of this in the experiments. Is the inclusion of L2 and l2 necessary, or have previous works discussed the necessity of incorporating these terms?

3. In Fig 2, if I understand correctly, the multimodal script in the top-left corner is used to generate a single keyframe. However, the three keyframes shown on the right under the VLM section could easily give the impression that they were all generated by the same script. This may lead to some confusion for readers. Additionally, according to Equation 1, the input to the decoder should also include a noise latent. However, the inference process shown at the bottom of Figure 2 does not fully display the complete inputs and outputs, which could lead to some misunderstandings.

**Questions:**

I have outlined some concerns and suggestions in the weaknesses section, and there are additional questions that need clarification.

1. Is the covariance matrix in Eq. 7 of the supplementary material a diagonal matrix?

2. In the few-shot training section, what is the “episode’s visual tokens” (line 302)?

3. In Fig 8, what are the reference images for generating the few-shot results?  If I understand correctly, face embedding is also used as part of the multimodal script to generate zero-shot results. For the few-shot generation, what images are used as references, and how do these reference images help improve identity consistency?

---

> ### Author Response · Authors · 2024-11-19
>
> We sincerely thank you for your thoughtful feedback and valuable suggestions!
>
> **W1. Typo of kd means and variances.**
>
> Thank you for pointing out the typo. The GMM parameters should be kd means and kd variances. We apologize for any confusion this may have caused.
>
> **W2. Experiments with Eq.2 only.**
>
> - We conduct additional experiments where only Eq. 2 is used as the objective function. The results indicate that relying solely on Eq. 2 causes the model to generate characters positioned near the image edges and, in some cases, even fail to generate the characters, which results in a significant decline in quality. While previous works have not specifically discussed the necessity of incorporating both L1 and L2, our experiments demonstrate that these two terms complement Eq. 2, resulting in higher-quality results. **In the appendix E.3 of the revised paper, we provide visualizations of the experimental results along with metrics for a detailed evaluation.**
>
> | Method  | CLIP↑ | Inception↑ | Aesthetic↑ | FID↓ | ST↑ | LT↑ |
> |:-------------:|:------------:|:------------:|:------------:|:------------:|:------------:|:------------:|
> | Con-L1-L2 | 19.584 | 9.698 | 6.093 | 2.043 | 0.646 | 0.814 |
> | Con-L1-L2-ref | **20.071** | **9.842** | **6.288** | **1.912** | **0.701** | **0.893** |
> | L1-L2 | 18.833 | 8.609 | 5.893 | 2.714 | 0.606 | 0.755 |
> | L1-L2-ref | 19.431 | 8.774 | 5.979 | 2.560 | 0.616 | 0.776 |
> | Con | 16.882 | 7.463 | 5.352 | 3.283 | 0.576 | 0.701 |
> | Con-ref | 17.039 | 7.580 | 5.411 | 3.136 | 0.571 | 0.708 |
>
> **W3. Confusion in Fig 2.**
>
> We sincerely appreciate your valuable suggestions and feedback. You are correct that the multimodal script in the top-left corner is used to generate a single keyframe. We will include clearer illustrations in the revised paper to enhance understanding.
>
>
> **Q1. Is covariance matrix diagonal?**
>
> Yes, the covariance matrix is diagonal.
>
> **Q2. What is the episode's visual tokens?**
>
> We apologize for the confusion. In this section, we randomly select 10 keyframes from the current video segment, encode them into visual tokens, and position them at the beginning of the input sequence. The term "episode's visual tokens" refers to these encoded keyframe tokens.
>
> **Q3. Reference images in Fig 8, and how do these images help?**
>
> In Fig. 8, we select 10 keyframes containing characters as references. These references improve the results because they serve as the reference example for in-context learning. More specifically, by providing some examples as context, the model can leverage these reference contexts to produce higher-quality outputs.
>
> Once again, we thank you for your constructive feedback! Please let us know if you have further suggestions or concerns!

---

> > ### Comment · Reviewer_4ibH · 2024-11-24
> >
> > Thank you for your response. Your clarifications have addressed most of my concerns; however, a few questions remain:
> >
> > 1. The revised paper does not mention Table 1 in the main text. Could you elaborate on its purpose and relevance?
> > 2. It is interesting that without L1 and L2, the generated characters tend to move toward the edges of the frame. Could you explain the potential reasons behind this behavior?
> > 3. If the characters described in the script are inconsistent with those provided in the few-shot generation references, which source will dominate the resulting generation?

---

> > > ### Author Response · Authors · 2024-11-25
> > >
> > > We sincerely appreciate your concerns and are delighted to have the opportunity to discuss them with you!
> > >
> > > **1. Correction on Table 1 Reference**
> > >
> > > - We apologize for mistakenly referencing Table 1 in our submission. We have uploaded a revised paper to address this issue, with the correction highlighted in red at L407.
> > > - Table 1 evaluates keyframes generated by different methods, encompassing metrics such as text alignment, keyframe quality, short-term (ST) and long-term (LT) consistency. The detailed computation of ST and LT consistency is also thoroughly explained in Appendix B.
> > >
> > > **2. Possible Reasons for Characters Being Positioned at the Image Edges**
> > >
> > > We believe that the continuous token supervision primarily focuses on modeling the overall distribution of  tokens, without explicitly accounting for layout coherence. In our setting, the overall distribution is primarily characterized by the distribution of characters. Therefore, under the supervision of continuous token supervision, the model is capable of generating higher-quality character IDs. On top of this, L1 and L2 further provide explicit supervision for each hidden state within the tokens, effectively imposing layout constraints.
> > >
> > > **3. Dominance of Current Script in Generation**
> > >
> > > - The current script is dominant. As the script for the current frame is the most recently provided input, the model prioritizes it.
> > >
> > > - **Few-shot references are merely supplementary and do not strictly enforce the model to follow them.** When inconsistencies arise, the model defaults to a zero-shot generation mode. This behavior stems from the training process of our autoregressive model, which allows the model to effectively utilize information that is genuinely useful for generating the current frame.
> > >
> > > - For example, when generating the 50th keyframe, the model has access to the script for the 50th frame, historical information from the previous 49 frames, and potentially few-shot references. During training, the model learns to select information that is helpful for the current frame generation, with a focus on the current script. Consequently, when consistent references are provided, the quality of the corresponding character improves as the model leverages more helpful information. On the other hand, inconsistent references do not negatively impact the generated results.
> > >
> > >
> > > Thank you once again for your valuable feedback and thoughtful comments! Please do not hesitate to share any further feedback, as we are always eager to learn and enhance the quality of our research.

---

> > > > ### Comment · Reviewer_4ibH · 2024-11-25
> > > >
> > > > Thank you for the author's patient response. Although most of my questions have been resolved and I am inclined to improve my score, I am particularly interested in the roles of L1 and L2 and have some remaining doubts. Since the author did not provide visualization results using only L1 + L2 in Fig. 22, it is difficult for me to infer their specific roles. Additionally, when using only the Continuous loss, why do the generated results tend to produce two characters, whereas adding L1 and L2 results in the generation of a single character? Is this due to differences in the input?

---

> > > > > ### Author Response · Authors · 2024-11-25
> > > > >
> > > > > Thank you for your feedback! We are delighted to have the opportunity to engage in this discussion with you.
> > > > >
> > > > > First, the visualization of results using only L1 + L2 loss is presented in Figure 6. From our understanding, you might be interested in comparing the outputs of models trained with different supervision under the same input conditions. To address this, **we have further provided additional visualizations in Figure 22 of the newly uploaded revised paper.** We believe this visualization will intuitively illustrate the role of different loss functions.
> > > > >
> > > > > - As shown in Figure 22, when using only L1 + L2 loss, the model learns a reasonable layout but produces results with poor character ID preservation.
> > > > > - On the other hand, when using only the continuous loss, although the overall quality of the generated results is poor, it is evident that the characters in the results are closer to the target ID.
> > > > > - Finally, when both the continuous loss and L1 + L2 loss are employed, they effectively fulfill their respective tasks, achieving a win-win situation.
> > > > >
> > > > > Regarding the observation that "the generated results tend to produce two characters," this is because the examples we showcased happened to be like this. The model trained with only the continuous loss fails to learn a reasonable layout, so "the generated results tend to produce two characters" is just one manifestation of this failure and does not represent the model's overall performance.
> > > > >
> > > > > Additionally, we have ensured the fairness of the comparative experiments. The training data and the input for generating all results are identical, with only the loss functions being different.
> > > > >
> > > > > We hope our response helps clarify your questions. If you have any further concerns, please don’t hesitate to let us know. We are more than happy to engage in such fruitful discussions! :)

---

> > > > > > ### Comment · Reviewer_4ibH · 2024-11-25
> > > > > >
> > > > > > Thank you for the author's response. My questions have been resolved, and I am happy to improve my score.

---

### Official Review · Reviewer_xVaX · 2024-11-04

**Soundness:** 3
**Presentation:** 3
**Contribution:** 2
**Rating:** 6
**Confidence:** 4

**Summary:**

This paper presents a hierarchical generation framework for long video sequence generation, which can be potentially applied to movie video generation. The key idea is to leverage the autoregressive models to generate key frame tokens, and then use diffusion models to decode the tokens into RGB frames. This approach leverages the advantage of any-length future prediction ability of AR model and the powerful rendering ability of diffusion models. Through some other techniques such as multimodal script, face embedding condition, etc., the framework can generate ultra-long video sequences with better quality.

**Strengths:**

+ The model provides an effective integration of autoregressive model and diffusion models, for the long visual sequence generation task. The combination is reasonable by successfully leveraging the arbitrary-length generation capability or AR model and the powerful rendering ability of diffusion models.
+ The utilization of the gaussian mixture model for the continuous visual token prediction is reasonable and inspiring.

**Weaknesses:**

- The paper proposes a lot of techniques to improve the identity consistency among different clips, however, I cannot find any ID-related evaluation on the generation results.
- The paper did not analyze the boundaries of the proposed method. For example,
    - when will the model fail at ID consistency since the model only provides one face embedding as the condition.
- The technical novelty is relatively limited. There exist many multimodal generative models can do the similar key frame generation task, though they may not be directly used for movie key generation.

**Questions:**

Please provide the evaluation of the ID consistency of the model, clarify the technical contributions, and discuss the limitations of the framework.

---

> ### Author Response · Authors · 2024-11-13
>
> Thank you for your valuable feedback and suggestions!
>
> **ID-consistency Evaluation**
>
> We have already provided ID-related evaluation in the paper, as seen in Table 1, with the ST and LT metrics. These are used to evaluate consistency in short-term and long-term, respectively. Detailed explanation for these two metrics is provided in Appendix B. Please let us know if you have any further questions.
>
> **Boundaries of Our Method**
>
> In Appendix F, we analyze the boundaries of the proposed method. Here, we summarize them as follows:
>
> - When too many characters appear simultaneously, the model may sometimes confuse character IDs. (Figure 22 of the original paper, or Figure 23 of the revised paper)
> - Since we perform sentence-level tokenization to ensure the resulting length, the model struggles to understand highly fine-grained and complex text prompts.
> - Our method does not control object-level IDs. This can be addressed by further constructing object-level IDs.
>
> **Technical Novelty and Contributions**
>
> Our method is not a trivial combination of existing approaches. Our motivation and intuition behind the design are fundamentally different from those of existing methods. We believe our technical novelty and contributions are as follows:
>
> - The problem we address is new. We solve an under-explored problem: unified long video generation and long story generation. To the best of our knowledge, few efforts have tackled this. Existing approaches are unable to generate videos of the length we achieve, and most are constrained by single scenes and lack controllability. Additionally, their results demonstrate significant drift over extended periods.
> - The motivation behind our autoregression + diffusion approach is completely different from prior works using autoregressive + diffusion models.
>   - Previous video generation methods mostly rely on diffusion models. While these methods achieve good results on short videos of a few seconds, they face fundamental challenges when generating long videos, especially those in the tens of minutes. Scaling up video diffusion model requires enormous computational resources and data, which is nearly impossible for ultra-long video tasks.
>   - In the image domain, previous works about autoregression + diffusion mainly focus on the instruction-following ability. However, our focus is on the autoregressive model's ability to model long sequences. Our innovative framework inherits the advantages of both diffusion and autoregressive models, achieving state-of-the-art long video generation for videos of tens of minutes.
> - Our approach unifies story generation and video generation in a hierarchical way.
> - Our method is orthogonal to existing video generation works. It can be integrated with various strong image-to-video models to produce higher-quality results.
> - Ensuring Length: We design compact yet representative compressed tokens to ensure both quality and context length. We further implement sentence-level tokenization, ensuring the generated results achieve the desired length despite the limitations of maximum sequence length.
> - Preserving ID: We implement an ID-preserving strategy to ensure the consistency of multiple characters.
> - Ensuring Controllability: We propose a multimodal script and innovatively implement sentence-level compression for text descriptions, ensuring both quality and extended context length. And we propose the corresponding data pipeline.
> - Supervision: Traditional autoregressive models are supervised with cross-entropy loss, but this is not feasible for our task, as we need to predict continuous tokens rather than discrete ones. Additionally, we observe poor performance when supervised solely with L1 and L2 loss. To address this, we supervise continuous real-valued token prediction, which effectively improves the model's performance.
> - We further explore the potential of our method, including few-shot in-context learning and implicit modeling of shot types.
> - New Metrics: We propose new ST and LT metrics to evaluate ID consistency from both short-term and long-term perspectives.
>
> In conclusion, our approach is grounded in a thorough and careful analysis of the strengths and weaknesses of autoregression and diffusion in video generation, rather than simply combining existing methods. The task we address is both novel and of significant practical relevance. Furthermore, we have explored various aspects of our method, such as shot types, which further demonstrates its value and potential.
>
> Once again, we are grateful for your thoughtful feedback! We hope these clarifications effectively address your concerns and we welcome any further discussion. Thank you for your time and attention.

---

> > ### Author Response · Authors · 2024-11-25
> > **Happy to provide additional clarification**
> >
> > Dear reviewer,
> >
> > Thank you for taking the time to review our paper!
> >
> > We understand that the review process can be time-consuming and we greatly appreciate your effort. As the response period is approaching, we want to know if there are any updates or feedback on our previous response. Please don’t hesitate to let us know if you have any further questions! :)

---

### Official Review · Reviewer_zniU · 2024-11-05

**Soundness:** 4
**Presentation:** 4
**Contribution:** 4
**Rating:** 8
**Confidence:** 4

**Summary:**

This paper presents MovieDreamer to make long video. This is challenging because current video diffusion models can only deal with short clips and do not have character consistency. The model trains a MLLM that can predicts the visual tokens with movie script conditioning. Then using the diffusion decoder to decode the condition into key frames, and using I2V to render it into video clips.

**Strengths:**

The quality is GREAT!
Engineering wise, this paper produces a product-level solution to long-video (movie) generation; Scientificly, this paper also explores training a MLLM to output visual features given the input scripts, instead of simply using an agent framework to compose scripts, and use subject-driven models to preserve character identity.
The ID Preserving is also very effective

**Weaknesses:**

Lack of non-human results, don't know if such methods can generalize well to general domain long-vedio generation.

**Questions:**

Since authors are changeing LLaMA input to CLIP embedding (for sentence), does it cause much for the LLM to adapt to such input?
How much cherry-pick do you need for a single long video?

---

> ### Author Response · Authors · 2024-11-13
>
> Thanks for your kind review on our paper!
>
> **Non-human Character Results**
>
> Thank you for your valuable feedback. As discussed in our Future Work, our ability to maintain ID consistency for non-human characters is limited. For example, when generating a dog, the results often differ from the specific target dog we aimed for. Therefore, our method currently excels when the main character is human.
>
> However, we have included this limitation in our future work. We believe that constructing and leveraging non-human IDs will effectively address this issue.
>
> **Adaptation Cost and Cherry-pick**
>
> During implementation, we observed that the LLM adapts well to sentence CLIP embedding inputs with minimal adaptation cost. We believe this is because:
> 1. Both the CLIP text encoder and LLaMA handle rich semantic features. CLIP sentence embeddings are highly semantic, and understanding these semantic embeddings is easy for LLaMA because of its strong language capabilities.
> 2. The structured multi-modal scripts further assist the model in effectively understanding the sentence embeddings.
>
> Our approach achieves a high success rate with minimal need for cherry-picking. Most of results are generated directly without selection.
>
> We hope these responses address your concerns. Please feel free to let us know if you have any further questions!

---

> > ### Comment · Reviewer_zniU · 2024-11-19
> >
> > Thank you for your rebuttal. I am happy to keep my supportive score.

---

### Author Response · Authors · 2024-11-19

First of all, we greatly appreciate all the reviewers' thoughtful comments!

**Our Technical Contribution**

We are delighted to see that our work is recognized by many reviewers. Here, we would like to take this opportunity to reiterate our contributions:

- Addressing an Under-Explored Task: We are the pioneer to tackle the significantly under-explored ultra-long video generation. Unlike most existing methods that extend videos through looping or simple motion, our approach fundamentally achieves greater temporal extension while maintaining consistency across multiple characters.

- Unify Story Generation and Video Generation: Our approach unifies story generation and video generation in a hierarchical way.

- Flexibility of Our Approach: Our method is compatible with any existing image-to-video generation method and benefits from state-of-the-art image-to-video methods. This further highlights the flexibility of our approach.

- Structured Multimodal Scripts: We design the structured multimodal scripts to enhance the controllability and introduce a multimodal scripts data pipeline tailored for long videos.

- Supervision: We employ continuous token supervision along with L1 and L2 losses to train the model and demonstrate that each of them contributes significantly to the model.

- Sentence-Level Tokenization: We implement sentence-level tokenization, ensuring the generated results achieve the desired length despite the limitations of maximum sequence length.

- Exploring Possibilities: We further explore the potential of our method, including few-shot in-context learning and implicit modeling of shot types.

- New Metrics: We propose new ST and LT metrics to evaluate video consistency from both short-term and long-term perspectives.

We believe these contributions demonstrate the novelty, robustness, and flexibility of our approach.

**Typos and Confusion**

We apologize for typos, grammar mistakes and unclear expressions.
We have carefully addressed the issues raised by the reviewers and made detailed revisions in the revised paper. **All revisions in the revised paper have been highlighted in red.**
Specifically, we:

- Improve Figure 2 to make it clearer and more comprehensible.
- Explicitly describe the losses introduced during DAE training for face embedding and description embedding.
- Clarify unclear expressions in the text, such as "episode's visual token."
- Correct typos, including "2kd means and 2kd variances."
- Add additional experiments to explore the importance of the L1 and L2 losses, further demonstrating that, L1 L2 together with the continuous token loss can jointly enhance model optimization.

Once again, we'd like to thank the reviewers for their careful readings and valuable comments. We welcome further discussion and are happy to make additional improvements!

---

> ### Author Response · Authors · 2024-11-24
>
> Dear reviewers,
>
> We sincerely appreciate your valuable efforts in reviewing our paper. As the Author-Reviewer discussion period is nearing its conclusion, we would like to kindly inquire if the reviewers have any additional questions or concerns regarding our previous responses. Please feel free to reach out if there are any remaining points that need clarification.
>
> Thank you!

---

### Meta-Review · Area_Chair_RgwF · 2024-12-17

**Metareview:**

The paper proposes a novel long-duration video generation framework, which is based on fine-tuning LLAVA to generate compressed tokens of key frames, which are then used to generate keyframes using a fine-tuned SDXL, and then videos using SVD. Extensive experiments demonstrate the effectiveness.

Initial reviews had mixed ratings 5686. Reviewers raised a number of concerns, including:
1. lack results of non-human videos (zniU)
2. any cherry picking done? (zniU)
3. missing evaluation on ID (xVaX)
4. boundary cases (limitations) not analyzed, e.g., will the model fail at ID consistency if one face embedding is provided (xVaX)
5. limited novelty (existing methods can do key-frame generation) (xVaX)
6. missing or unclear descriptions about method (4ibH, Jjeo)
7. missing ablation study on objective function (4ibH)
8. training cost? (Jjeo)
9. still obvious gap for target ID evaluation (Jjeo).
10. need to use video generation metrics, not just image-level (Jjeo).

The authors provided a response to address these concerns, including providing ablation studies, discussing limitations, and expanding on the novelty.  During discussion, authors clarified many points with 4ibH and added more interpretations of the various results / ablations. In the end, all reviewers were satisfied after the rebuttal and discussion, and final reviews were all positive 6688.  Reviewers appreciated the quality of the result, as well as the unique problem addressed.

Authors should update the paper based on the reviews, response, and discussions.

**Additional Comments On Reviewer Discussion:**

see above

---

### Decision · Program_Chairs · 2025-01-22

Accept (Poster)